# Accurate Elucidation of Oxidation Under Heavy Ozone Pollution: A Full Suite of Radical Measurement In the Chemical-complex Atmosphere

Renzhi Hu[1], Guoxian Zhang[1,2,×], Haotian Cai[1], Jingyi Guo[1], Keding Lu[4], Xin Li[4], Shengrong Lou[5], Zhaofeng Tan[4], Changjin Hu[1], Pinhua Xie[1,3, ××], Wenqing Liu[1,3]

[1] Key Laboratory of Environment Optics and Technology, Anhui Institute of Optics and Fine Mechanics, HFIPS, Chinese Academy of Sciences, Hefei, China

[2] School of Physics and New Energy, Xuzhou University of Technology, Xuzhou, China

[3] College of Resources and Environment, University of Chinese Academy of Science, Beijing, China

[4] State Key Joint Laboratory of Environmental Simulation and Pollution Control, College of Environmental Sciences and Engineering, Peking University, Beijing, China

[5] State Environmental Protection Key Laboratory of the Formation and Prevention of Urban Air Pollution Complex, Shanghai Academy of Environmental Sciences, Shanghai, China

×**Correspondence to:** Guoxian Zhang, School of Physics and New Energy, Xuzhou University of Technology, Xuzhou, China

××**Correspondence to:** Pinhua Xie, Key Laboratory of Environment Optics and Technology, Anhui Institute of Optics and Fine Mechanics, HFIPS, Chinese Academy of Sciences, Hefei, China

**Email addresses:** gxzhang@aiofm.ac.cn (Guoxian Zhang); phxie@aiofm.ac.cn (Pinhua Xie)

**Abstract:** The Yangze River Delta (YRD) in China encountered with prolonged ozone pollution in September 2020, which had significant impacts on the respiratory, dermatological, and visual health of local residents. To accurately elucidate the limitations of oxidation processes in the chemical-complex atmosphere, a full suite of radical measurements (OH, $HO_2$, $RO_2$, and $k_{OH}$) was established in YRD region for the first time. The diurnal peaks of radicals exhibited considerable variation due to environmental factors, showing ranges of 3.6 to $27.1 \times 10^6$ $cm^{-3}$ for OH, 2.1 to $33.2 \times 10^8$ $cm^{-3}$ for $HO_2$, and 4.9 to $30.5 \times 10^8$ $cm^{-3}$ for $RO_2$. At a heavy ozone pollution episode, the oxidation capacity reached an intensive level compared with other sites, and the simulated OH, $HO_2$, and $RO_2$ radicals provided by the RACM2-LIM1 mechanism failed to adequately match the observed data both in radical concentration and experimental budget analysis. Sensitivity tests utilizing a comprehensive set of radical measurements revealed that the higher aldehyde mechanism (HAM) effectively complements the non-traditional regeneration of OH radicals, yielding enhancements of 4.4% - 6.0% compared to the base scenario, while the concentrations of $HO_2$ and $RO_2$ radicals have shown increments of about 7.4% and 12.5%, respectively. Notably, $RO_2$ radical concentrations exhibit a pronounced sensitivity to autoxidation, with the incorporation of additional OVOCs potentially boosting simulated $RO_2$ radical concentrations by 20% to 40%. The incorporation of larger alkoxy radicals stemming from monoterpenes has refined the consistency between measurements and modeling in the context of ozone production under elevated NO levels, diminishing the disparity from 4.17 to 2.33. This outcome corroborates the hypothesis of sensitivity analysis as it pertains to ozone formation. Moving forward, by implementing a comprehensive radical detection approach, further investigations should concentrate on a broader range of OVOCs to rectify the imbalance associated with $RO_2$ radicals, thereby providing a more precise understanding of oxidation processes during severe ozone pollution episodes.

**Keywords:** FAGE-LIF; Full-chain detection; Radical; P(Ox); OVOCs;

# 1 Introduction

In recent years, China's rapid economic development has led to severe environmental pollution problems, which significant impacted the respiratory, dermatological, and visual health of local residents (Wang et al., 2022c; Huang et al., 2018). This has raised concerns about the coexistence of regional primary and secondary pollution, making air quality improvement efforts a focal point (Liu et al., 2021; Wang et al., 2022a). In the complex atmosphere, near-surface ozone ($O_3$) is formed through continuous photochemical reactions between nitrogen oxides ($NOx \equiv NO+NO_2$) and volatile organic compounds (VOCs) under light conditions, while hydroxyl radicals (OH) serve as the main oxidant in the troposphere, converting VOCs into hydroperoxy ($HO_2$) and organic peroxy ($RO_2$) radicals (Rohrer et al., 2014; Hofzumahaus et al., 2009). Additionally, the oxidation of nitric oxide (NO) and peroxy radicals produce nitrogen dioxide ($NO_2$), which is the sole photochemical source of ozone (Lu et al., 2012; Stone et al., 2012).

Despite numerous experimental and theoretical explorations to establish the radical-cored photooxidation mechanism in the troposphere, field observations were primarily focused on HOx ($HOx \equiv OH + HO_2$) radicals due to the limitations of detection technology (Kanaya et al., 2012; Lu et al., 2012; Hofzumahaus et al., 2009; Yugo Kanaya and Tanimoto, 2007; Ren et al., 2008; Stone et al., 2012; Levy, 1971). Recent advancements in detection technology, such as the application of a new LIF technique (ROxLIF), have made the detection of $RO_2$ radicals possible (Whalley et al., 2013; Tan et al., 2017a). Moreover, the union of comprehensive field campaigns and box model, has proven to be an effective method for verifying the integrity of radical chemistry at local to global scales (Lu et al., 2019b; Tan et al., 2018). Several experiments have indicated that the existing atmospheric chemical mechanism posted challenges in deepening the understanding of the regional pollution explosion (Whalley et al., 2021; Slater et al., 2020; Tan et al., 2017a; Woodward-Massey et al., 2023). For instance, the observation of up to $4 \times 10^9$ cm$^{-3}$ of $RO_2$ radical in the center of Beijing in 2017 (APHH) was significantly underestimated by the MCM v3.3.1 mechanism (Whalley et al., 2021). Further exploring the unreproducible concentration and the oxidation process in the chemical-complex atmosphere is deemed necessary (Whalley et al., 2021; Woodward-Massey et al., 2023).

The YRD region, situated between the North China Plain (NCP) and Pearl River
Delta (PRD), is highly prone to regional transport interactions and aerosol-boundary layer
feedback (Jia et al., 2021; Huang et al., 2020). In September 2020, the YRD region
experienced a severe episode of secondary pollution, with both the daily maximum
8-hour average ozone (MDA8) and daily average $PM_{2.5}$ concentrations surpassing the
pollution threshold, distinguishing it from other megacities (Fig. S1). In an effort to gain
a better understanding between the complex radical chemistry and the intensive oxidation
level, TROPSTECT-YRD (The experiment on Radical chemistry and Ozone Pollution
perSpectively: long-Term Elucidation of the photochemiCal oxidaTion in the Yangze
River Delta) was conducted in Hefei during September 2020. Accurate elucidation of the
oxidation process under heavy ozone pollution was provided by a full suite of radical
measurement (OH, $HO_2$, $RO_2$ and $k_{OH}$) in the chemical-complex atmosphere.

## 2 Materials and methods

### 2.1 Site description and instrumentation

The TROPSTECT observation was conducted from 1 to 20 September 2020 at the
Science Island background station (31.9° N, 117.2°E) in Hefei, a typical megacity located
in the central region of Anhui Province within the Yangtze River Delta. The station is
situated on a peninsula with abundant vegetation to the northwest of urban areas and is in
close proximity to Dongpu Lake, which is only 200 meters away, and the main road,
positioned 250 meters southward (Fig. 1). Consequently, the relatively enclosed
environment exhibits typical suburban characteristics of anthropogenic emissions. The
station is located in the transition region between urban and suburban areas, reflecting the
regional transpor of pollution in Hefei and its surrounding areas.

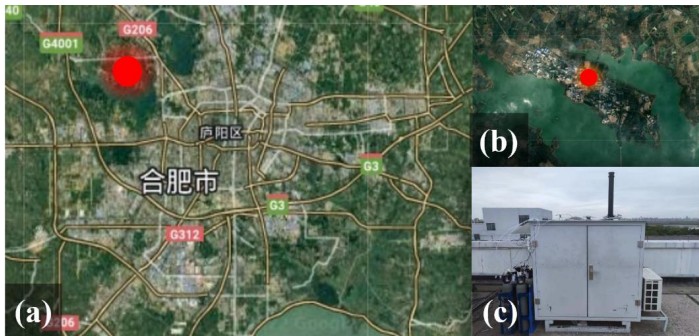

**Fig. 1. (a)** The location of the measurement site (source: © Google Earth).
**(b)** The close shot of the measurement site location (source: © Google Earth).
**(c)** The actual image for the LIF-Box.

Regarding the instrumentation, a group of oxidation-related instruments were
installed on the 7th floor of the Comprehensive Building at the Anhui Institute of Optics
and Fine Mechanics (AIOFM), with all sampling outlets positioned more than 20 meters
above the ground. The details of the instruments measuring various parameters such as
meteorological factors (WS, WD, T, RH, P, Jvalues), gas pollutants ($O_3$, CO, $SO_2$, NO,
$NO_2$, HONO, HCHO, PAN), and non-methane hydrocarbons (NMHCs) are provided in
Table S1.
The measurement of NO, $NO_2$, $O_3$, CO, and $SO_2$ was carried out using commercial
Thermo Electron model series instruments. Thereof, NO and $NO_2$ were measured using a
chemical fluorescence method (CL) with an enhanced trace-level NO-$NO_2$-NOx analyzer
(PKU-PL), which achieved a detection limit of 50 ppt (Tan et al., 2017a). The detection
of $O_3$ and $SO_2$ was conducted through Thermo Electron model 49i and 43i, respectively,
while Thermo Electron model 48i was utilized for CO detection. Cavity ring-down
spectroscopy (CRDS, Picarro-G2401) was employed for CO detection in parallel, and
another ultraviolet absorption instrument (Ecotech EC9810B) was for ozone detection.
The instrument inlets were placed within 5 meters of each other for comparison.
To ensure measurement accuracy, the instruments in the campaign underwent zero
point calibration procedures during the early (August 31st) and late (September 21st)
observation periods, and cross-calibrations for $O_3$ and CO measurements were carried out
during the middle (September 9th). Furthermore, additional zero calibration for Thermo
48i CO detection was undertaken daily from 0:00-0:30 to minimize shift correction. The
comparison results revealed high consistency within the instrument accuracy range for
both CO and $O_3$ measurements (Fig. S2(a)(b)).
HONO was detected using a home-built instrument by cavity-enhanced absorption
spectroscopy (CEAS), while formaldehyde was determined by the Hantzsch method
(SDL MODEL 4050) (Duan et al., 2018; Yang et al., 2021a). An automated gas
chromatograph equipped with a mass spectrometer and flame ionization detector
(GC-FID/MS) was employed for the online measurement of 99 VOCs species.
Information table for parts of the VOC monitoring species by online GC-MS/FID was
listed in Table S2.
The eight crucial photolysis frequencies (j($NO_2$), j($H_2O_2$), j(HCHO_M), j(HCHO_R),
j(HONO), j(NO$_3$_M), j(NO$_3$_R), j(O$^1$D)) were directly measured by a photolysis
spectrometer (Metcon, Germany). The unmeasured photolysis frequencies of the
remaining active species were computed using Eq.(1):

$$j = l \cdot cos(\chi)^m \cdot e^{-n \cdot sec(\chi)} \qquad (1)$$

The variations in photolysis frequency due to solar zenith angle ($\chi$) were adjusted based
on the ratio of observed and simulated j(NO$_2$). The optimal values for parameters ($l$, $m$,
and $n$) for different photolysis frequencies were extensively detailed by the MCM v3.3.1
(http://mcm.york.ac.uk/parameters/photolysis_param.htt) (Jenkin et al., 2003; Jenkin et
al., 1997).

## 2.2 Radical measurement

### 2.2.1 OH, HO$_2$, RO$_2$ Concentrations

The laser-induced fluorescence instrument developed by the Anhui Institute of Optics
and Fine Mechanics (AIOFM-LIF) was used to simultaneously detect the concentrations
of OH, HO$_2$, and RO$_2$ radicals, along with OH reactivity ($k_{OH}$). The OH radical was
directly measured by detecting on-resonance fluorescence excited by a 308 nm laser. An
indirect measurement for HO$_2$ was carried out after converting it to OH at a fixed
efficiency (Heard and Pilling, 2003).
The laser utilized for fluorescence excitation is a high-frequency tunable dye laser
that emits a 308 nm laser, with the laser power divided into a ratio of 0.45:0.45:0.08:0.02.
Of this power, 90% is directed towards fluorescence cells for concentration and reactivity
detection via optical fibers, respectively. 8% of the laser power is directed to a reference
cell, while the remaining 2% is used to monitor real-time power fluctuations. The laser is
transmitted through HO$_2$, OH, and RO$_2$ cells in turn via a coaxial optical path. Two
photodiodes are set up at the end of the reference cell and RO$_2$ detection cell, respectively.
The voltage signals and power fluctuations are compared synchronously to diagnose the
laser stability. To maintain detection efficiency, the power inside the measurement cells
should not be less than 10 mW. Sampling nozzles of 0.4 mm are deployed above OH and
HO$_2$ cells for efficient sampling at a flow rate of approximately 1.1 SLM, and the
pressure for all fluorescence cells are maintained at 400 Pa. The micro-channel plate
(MCP) detects the weak fluorescence signal collected by lens systems with low noise and
high gain. Additionally, a digital delay generator (DG645) optimizes the timing control

between the laser output, signal detection, and data acquisition. All of these modules are integrated into a sampling box with constant air conditioning, except for the laser.

The detection of $RO_2$ radicals is more complex compared to the integrated detection of OH and $HO_2$ radicals (Whalley et al., 2013). To achieve the complete chemical conversion from ROx to $HO_2$, a crucial role is played by a 66 mm×830 mm aluminium flow tube, whose performance has been confirmed through the CHOOSE-2019 field campaign (Li et al., 2020). A mixture of 0.17% CO and 0.7 ppm NO injected into the flow tube facilitates the reduction of heterogeneous radical loss and enhancement of conversion efficiency. The sampling flow is limited to 7 SLM by a 1 mm nozzle, and the tube pressure is maintained at 25 hPa. In contrast to the HOx cells, the large-diameter nozzle (4 mm) is equipped above the cell, and a high concentration of NO (~300 ppm) facilitates the full magnitude $HO_2 \rightarrow OH$ conversion.

The observation data ($H_2O$, $O_3$) is combined with experimental characterization to eliminate ozone photolysis interference, and most interference signals are excluded by utilizing wavelength modulation (Zhang et al., 2022a). A comparison experiment with PKU-LIF demonstrated the consistency of OH measurement in complex atmosphere (Zhang et al., 2022b). An additional atmospheric oxidation observation was conducted in the same location and season in 2022 with a chemical modulation method to determine the chemical background of OH radicals (Fig. S3). During the ozone pollution (2022.9.29-2022.10.3), the daytime peaks of ozone concentration above 75 ppb, accompanied by alkene species approaching ~10 ppb. The diurnal concentration of isoprene was also a high level (>1 ppb). The chemical conditions are more favourable to induce OH interference than in the TROPSTECT campaign, while the OH concentrations achieved by chemical modulation ($OH_{chem}$) and wavelength modulation ($OH_{wav}$) were in good agreement. No obvious chemical background was observed by deploying an inlet pre-injector. Therefore, it is not expected that OH measurement in the present study was affected by internal interference.

For $HO_2$ measurement, lower NO concentration (~$1.6 \times 10^{12}$ cm$^{-3}$, corresponding to ~15% conversion efficiency) are selected to limit the $RO_2 \rightarrow HO_2$ interference to less than 5% (Wang et al., 2021). Since only the total-$RO_2$ mode is used for the campaign, the additional uncertainty introduced by $RO_2/R(OH)O_2$ classification is negligible (Tan et al.,

2017b). The observed maximum daily PAN (11:00-14:00) is only 1.15±0.67 ppb,
resulting in a calculated PAN-pyrolytic interference for $RO_2$ measurement of less than 1
ppt (Fuchs et al., 2008). The general applicability of AIOFM-LIF in complex atmosphere
has been demonstrated through various campaigns (Zhang et al., 2022b; Wang et al.,
2021; Wang et al., 2019a).
To complete the calibration task, a standard source stably generates equal amounts
of OH and $HO_2$ radicals (Wang et al., 2020). The radical source is also capable of
yielding specific $RO_2$ by titrating hydrocarbon with OH. It is noteworthy that $CH_3O_2$ has
the highest mixing ratio in the $RO_2$ species, thus it was chosen to represent for sensitivity
calibration. The instrument is calibrated every two days, except during rainy weather. The
limit of detection (LOD) for OH, $HO_2$, and $RO_2$ in different cells with a typical laser
power of 10 mW is measured at $3.3×10^5$ $cm^{-3}$, $1.1×10^6$ $cm^{-3}$, and $2.5×10^6$ $cm^{-3}$,
respectively (60 s, 1σ). Measurement accuracy for OH, $HO_2$, and $RO_2$ radicals are
reported to be 13%, 17%, and 21%, respectively.
**2.2.2 OH reactivity($k_{OH}$)**
The detection of $k_{OH}$ in the atmosphere, defined as the reciprocal of OH lifetime, was
conducted using a laser flash photolysis laser-induced fluorescence (LP-LIF) instrument
(Lou et al., 2010). The configuration structure for $k_{OH}$ measurement has been detailed in a
previous study(Liu et al., 2019). The flow tube in the OH production-reaction unit is at
ambient pressure, with a gas flow rate of 17 SLM. A pulsed laser beam (266 nm with an
average power of 15 mJ) is output from a frequency-quadrupled Nd:YAG laser, which
generates stable OH radical through flash photolysis of ambient ozone in the flow tube.
Consistent and stable production of OH radicals is ensured by maintaining a stable
concentration of reactants, flow field, and laser energy. Under conditions of 80 ppb $O_3$
and 8000 ppm water vapor concentration, OH radicals produced in the flow tube remains
at the concentration order of $10^9$ $cm^{-3}$. Subsequently, the OH radicals are sampled through
a nozzle into a fluorescence cell. The OH fluorescence signal is then detected using laser
pump and probe techniques and is fitted to calculate the slope of OH decay ($k_{OH}$). The
detection accuracy, achieved with an integration time of 180 s, is 0.3 $s^{-1}$ (1σ).
**2.3 Observation-Based Model**
The Regional Atmospheric Chemical Mechanism version 2 (RACM2) incorporating

the latest Leuven isoprene mechanism (LIM) was utilized to simulate the concentrations and reactions of OH, $HO_2$, and $RO_2$ radicals (Stockwell et al., 1997; Griffith et al., 2013; Peeters et al., 2014). The RACM2-LIM1 mechanism was specifically involved with fewer species compared to the explicit MCM mechanism, thus ensuring higher operational efficiency (Liu et al., 2022). The comprehensive list of model constraints was provided in Table S3. The measured NMHCs include 29 alkanes, 11 alkenes, 15 aromatics, as well as acetylene and isoprene. For the base scenario, boundary conditions were established using the observed species, with assumed concentrations of hydrogen ($H_2$) and methane ($CH_4$) at 550 ppb and 1900 ppb, respectively. An ozone-simulation test was conducted to determine the suitable atmospheric lifetime ($\tau_D$) for the base model. At the lifetime of 24 hours, with a corresponding first-order loss rate of 1.1 cm/s (assuming a boundary layer height of 1 km), the simulated ozone concentration closely matched the observed values (Fig. S4). To improve the model-measurement consistency between OH, $HO_2$ and $RO_2$ radicals, a series of sensitivity analyses were performed to evaluate the impacts of potential mechanisms, as detailed in Table 1. The time resolution of all constraints was uniformly set to 15 minutes through averaging or linear interpolation. To reinitialize unconstrained species to a steady-state, three days of data were input in advance as the spin-up time.

**Table.1.** The sensitive test scenarios utilized to improve the model-measurement consistency between OH, $HO_2$ and $RO_2$ radicals.

| Scenario | Configuration | Purpose |
|---|---|---|
| Base | RACM2 updated with isoprene reaction scheme (LIM) | The base case with the species involved in Table S3 are constrained as boundary conditions. |
| X on | As the base scenario, but add the X mechanism, and the X level is between 0.25 - 0.5 ppb. | To untangle the missing OH source where base scenario failed. |
| MTS on | As the base scenario, but add a monoterpene source, and the monoterpene level is ~0.4 ppb. | Utilizing monoterpene-derived $RO_2$ to represent the alkoxy radicals with rather complex chemical structures. |
| MTS+X on | As the base scenario, but both the X mechanism and monoterpene source are considered. | To consider both the missing OH and $RO_2$ sources. |
| HAM on | As the base scenario, but add the reactive aldehyde chemistry. | To provide a test of whether the proposed mechanism can explain the missing OH source. |
| HAM on (4 × ALD) | As the base scenario, but add the reactive aldehyde chemistry, and the concentration of ALD was amplified by a factor of 4. | To quantify the impact of missing aldehyde primary emissions on ROx chemistry. |
| Ozone simulation | As the base scenario, but remove the constraints of the observed ozone and NO concentrations. | To test the suitable lifetime for the base model. |
| HCHO simulation | As the base scenario, but remove the constraint of the observed HCHO concentration. | To test the simulation effect of the existing mechanism on formaldehyde |

The local formation of ozone can be accurately quantified through the online measurement of ROx radicals (Tan et al., 2018). To overcome the interference from NO, the total oxidant (Ox), which is defined as the sum of $NO_2$ and $O_3$, can serve as a reliable parameter to indicate the level of oxidation. Eq.(2) shows that the rate of NO oxidation by peroxy radicals is equivalent to the production of $O_3$, denoted as F(Ox):

$$F(O_x) = k_{HO_2+NO}[NO][HO_2] + \sum_i k_{RO_2^i+NO}[NO]RO_2^i \tag{2}$$

The major loss pathways for Ox encompass ozone photolysis, ozonolysis reactions, and radical-related reactions ($OH/HO_2+O_3$, $OH+NO_2$), represented as D(Ox) in Eq.(3):

$$D(O_x) = \varphi_{OH}j(O^1D)[O_3] + \Sigma i \left\{\varphi_{OH}^i k_{Alkenes+O_3}^i [Alkenes][O_3]\right\} + (k_{O_3+OH}[OH] + k_{O_3+HO_2}[HO_2])[O_3] + k_{OH+NO_2}[OH][NO_2] \tag{3}$$

Here, the OH yields from ozone photolysis and ozonolysis reactions are denoted as $\varphi_{OH}$ and $\varphi_{OH}^i$, respectively.

The net photochemical Ox production rate in the troposphere, denoted as P(Ox) in Eq.(4), can therefore be calculated as the difference between Eqs. (2) and (3):

$$P(O_x) = F(O_x) - D(O_x) \tag{4}$$

## 2.4 Experimental budget analysis

In this study, an experimental radical budget analysis was also conducted (Eqs. (5) - (12)). Unlike model studies, this method relies solely on field measurements (concentrations and photolysis rates) and chemical kinetic data, without depending on concentrations calculated by models(Whalley et al., 2021; Tan et al., 2019b). Given the short-lived characteristics of OH, $HO_2$, and $RO_2$ radicals, it is expected that the concentrations are in a steady state, with total production and loss rates being balanced(Lu et al., 2019a). By comparing the known sources and sinks for radicals, unknown processes for initiation, transformation and termination can be determined.

$$P(OH) = j_{HONO}[HONO] + \varphi_{OH}j(O^1D)[O_3] + \Sigma i \left\{\varphi_{OH}^i k_{Alkenes+O_3}^i [Alkenes][O_3]\right\} + (k_{HO_2+NO}[NO] + k_{HO_2+O_3}[O_3])[HO_2] \tag{5}$$

$$D(OH) = [OH] \times k_{OH} \tag{6}$$

$$P(HO_2) = 2 \times j_{HCHO\_R}[HCHO] + \Sigma i \left\{ \varphi_{HO_2}^i k_{Alkenes+O_3}^i [Alkenes][O_3] \right\}$$
$$+ (k_{HCHO+OH}[HCHO] + k_{CO+OH}[CO])[OH]$$
$$+ \alpha k_{RO_2+NO}[NO][RO_2] \tag{7}$$

$$D(HO_2) = (k_{HO_2+NO}[NO] + k_{HO_2+O_3}[O_3] + k_{HO_2+RO_2}[RO_2]$$
$$+ 2 \times k_{HO_2+HO_2}[HO_2])[HO_2] \tag{8}$$

$$P(RO_2) = \Sigma i \left\{ \varphi_{RO_2}^i k_{Alkenes+O_3}^i [Alkenes][O_3] \right\}$$
$$+ k_{OH}[VOCs][OH] \tag{9}$$

$$D(RO_2) = \left\{ (\alpha + \beta)k_{RO_2+NO}[NO] + (2 \times k_{RO_2+RO_2}[RO_2] \right.$$
$$\left. + k_{HO_2+RO_2}[HO_2]) \right\}[RO_2] \tag{10}$$

$$P(RO_x) = \Sigma i \left\{ (\varphi_{OH}^i + \varphi_{HO_2}^i + \varphi_{RO_2}^i)k_{Alkenes+O_3}^i [Alkenes][O_3] \right\} + j_{HONO}[HONO]$$
$$+ \varphi_{OH}j(O^1D)[O_3] + 2 \times j_{HCHO\_R}[HCHO] \tag{11}$$

$$D(RO_x) = (k_{OH+NO_2}[NO_2] + k_{OH+NO}[NO])[OH] + \beta k_{RO_2+NO}[NO]$$
$$+ 2 \times (k_{RO_2+RO_2}[RO_2][RO_2] + k_{HO_2+RO_2}[HO_2][RO_2]$$
$$+ k_{HO_2+HO_2}[HO_2][HO_2]) \tag{12}$$

In which, j(HONO), j(O$^1$D) are the measured photolysis rates of HONO and O$_3$, respectively, and jHCHO_R is the measured photolysis rate for the channel of formaldehyde photolysis generating HO$_2$. $\varphi_{OH}$ represent the OH yield in the O$_3$ photolysis reaction. $\varphi_{OH}^i$, $\varphi_{HO_2}^i$ and $\varphi_{RO_2}^i$ are the yields for the ozonolysis reaction producing OH, HO$_2$, and RO$_2$, respectively. α is the proportion of RO$_2$ radicals reacting with NO that are converted to HO$_2$, and β is the proportion of alkyl nitrates formation, which are set to 1 and 0.05, respectively(Tan et al., 2019b).

# 3 Results

## 3.1 Overview of Measurement

During the observation period, the meteorological parameters and trace gas concentrations were plotted in Fig. S5. The timeseries revealed that the peak temperature exceeded 30°C, and the humidity levels remained between 30 – 50% during the daytime. The photolysis rates were observed to peak at noon (11:00 – 13:00), with j(O$^1$D) and j(NO$_2$) reaching approximately $3\times10^{-5}$ s$^{-1}$ and $8\times10^{-3}$ s$^{-1}$, respectively. Brief rainfall events temporarily happened on September 10th, 15th, and 17th, but totally favorable meteorologies induced the prolonged ozone pollution. The daily maximum 8-hour

average ozone concentration (MDA8), as depicted in Fig. 2, consistently exceeded the Chinese Grade I national air quality standard (GB3095-2012) throughout the observation, with nine days exceeding the Grade II standard.

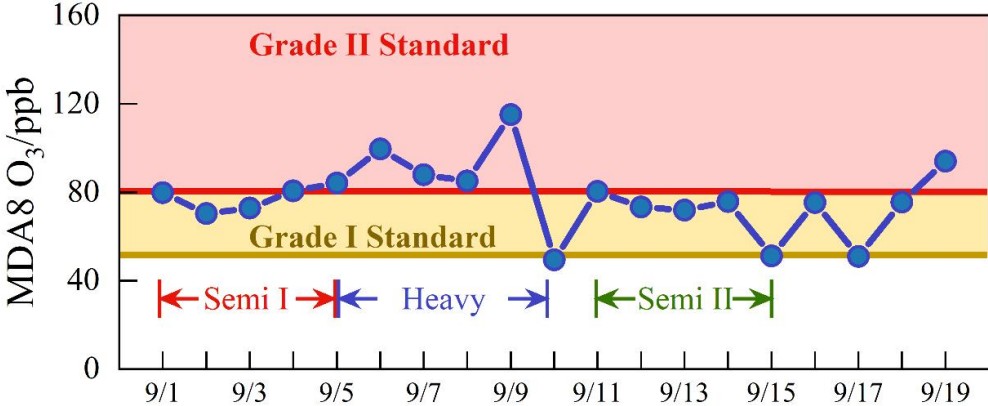

**Fig. 2.** The daily maximum 8 h average $O_3$ during the campaign. The yellow and red lines denote the Grade I and Grade II national standards for $O_3$, respectively. Brief rainfall events temporarily happened on10, 15, and 17 Sep.

The ozone pollution can be categorized into three continuous periods based on pollution levels, which disclose transitional 'Semi - Heavy - Semi' pollution characteristics. Fig. 3 depicts daily variations in meteorological and trace gas concentrations for different periods. During the Semi I (1 to 5 September) and Semi II (11 to 14 September) periods, the MDA8 levels exceeded Grade I standard, with an average value of 75.92±5.14 ppb and 75.45±3.73 ppb, respectively. Notably, NO levels peaked around 9:00 and rapidly decreased to a few hundred ppt due to photochemistry. In addition, HONO and $NO_2$ exhibited bimodal variations, with diurnal concentration ranges of 0.09 – 0.50 ppb and 3.35 – 13.77 ppb, respectively. The HONO/$NO_2$ ratios during both Semi periods were consistent with previous urban/suburban observations, with daytime values of 0.049±0.014 and 0.035±0.012, respectively (Yang et al., 2021b; Shi et al., 2020; Hu et al., 2022). Isoprene levels accumulated during the day and decreased at night during both Semi pollution episodes, with a diurnal average concentration in Semi II only 49.3% of that in Semi I (0.71±0.087 ppb vs 0.35±0.073 ppb). Formaldehyde, as the key oxidation species, exhibited a concentration profile mirroring that of isoprene, with significantly higher concentrations ranging from 1.20 to 36.34 ppb compared to other urban regions (Ma et al., 2022; Yang et al., 2022; Tan et al., 2017b; Yang et al., 2021a). Heavy pollution episodes from 5 to 9 September resulted in daytime ozone concentration as high as 129.9 ppb, and oxidation-related species such as HCHO, HONO, NOx, and

VOCs increased synchronously compared to other days.

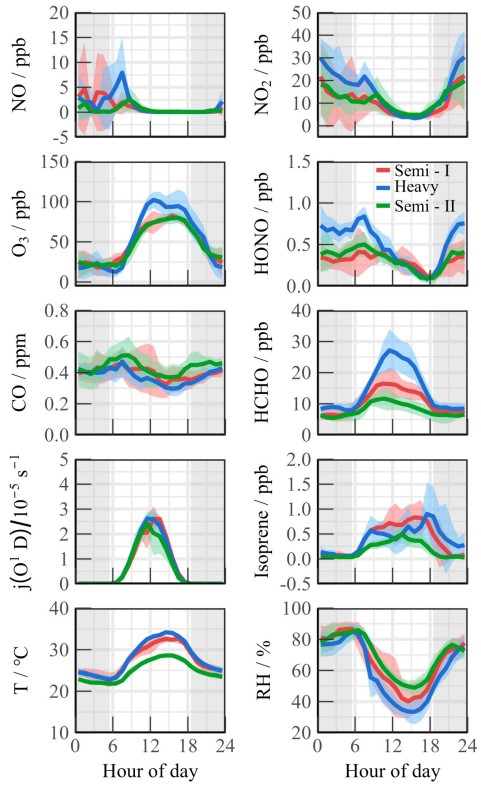

**Fig. 3.** Mean diurnal profiles of observed meteorological and chemical parameters during the campaign. Three periods
were divided for subsequent study (Semi I, Heavy, and Semi II).

## 3.2 ROx radical concentrations and budgets

The observed and modeled timeseries for OH, HO$_2$, RO$_2$, and $k_{OH}$ during the observation time are depicted in Fig. 4. The diurnal peaks of radicals exhibited a wide span due to changes in environmental conditions, with ranges of $3.6 - 27.1 \times 10^6$ cm$^{-3}$ for OH, $2.1 - 33.2 \times 10^8$ cm$^{-3}$ for HO$_2$, and $4.9 - 30.5 \times 10^8$ cm$^{-3}$ for RO$_2$. Continuous data for $k_{OH}$ observation were acquired within a range of $8.6 - 30.2$ s$^{-1}$. Fig. S6 presents the diurnal profiles of the observed and modeled values during different episodes. The diurnal maximum of OH radical at noon differed between Semi I and Semi II, with $9.28 \times 10^6$ cm$^{-3}$ and $5.08 \times 10^6$ cm$^{-3}$, respectively, while total peroxy radicals (HO$_2$+RO$_2$) remained at similar levels with $19.43 \times 10^8$ cm$^{-3}$ and $18.38 \times 10^8$ cm$^{-3}$. Additionally, the distribution of peroxy radicals are not similar in the two Semi periods, with HO$_2$/RO$_2$ ratios of 1.69:1 and 0.76:1, respectively, which reflects the uneven oxidation levels between Semi I and Semi II. During the Heavy ozone pollution, the averaged OH, HO$_2$, and RO$_2$ concentrations were 1.90, 2.15, and 1.98 times higher than those in the Semi

periods, suggesting a stronger oxidation capacity, with $k_{OH}$ in Heavy being 26.43% and 9.56% higher than in Semi I and Semi II, respectively. Limited anthropogenic emissions in the suburban environment reduced the oxidation contribution by NOx and CO (27.59%). During the heavy pollution, organic species exhibited dominant behavior regarding diurnal reactivity (9.22 s$^{-1}$ for 69.79%), and anthropogenic hydrocarbons were not major $k_{OH}$ sources. With an abundant level (~1 ppb), isoprene contributed more than 10% of the reactivity in the diurnal cycle. Therefore, the effect of BVOCs species (such as monoterpenes, limonene, etc.) on radical chemistry cannot be ignored (Ma et al., 2022; Wang et al., 2022b). $k_{OVOCs}$ are categorized into three groups: $k_{OVOCs(Obs)}$, $k_{OVOCs(Model)}$, and $k_{HCHO}$. Given the significance of formaldehyde photolysis, the contribution of HCHO to $k_{OVOCs}$ is distinguished. $k_{OVOCs(Obs)}$ encompasses species observed in addition to formaldehyde, such as acetaldehyde (ACD) and the oxidation products of isoprene (MACR and MVK). Intermediates generated by the model, including glyoxal (GLY), methylglyoxal (MGLY), higher aldehydes (ALD), ketones (KET), methyl ethyl ketone (MEK), and methanol (MOH), are classified as $k_{OVOCs(Model)}$. Upon considering $k_{OVOCs(Model)}$, the reactivity calculated prior to September 10th aligns quite well with the observed OH reactivity.

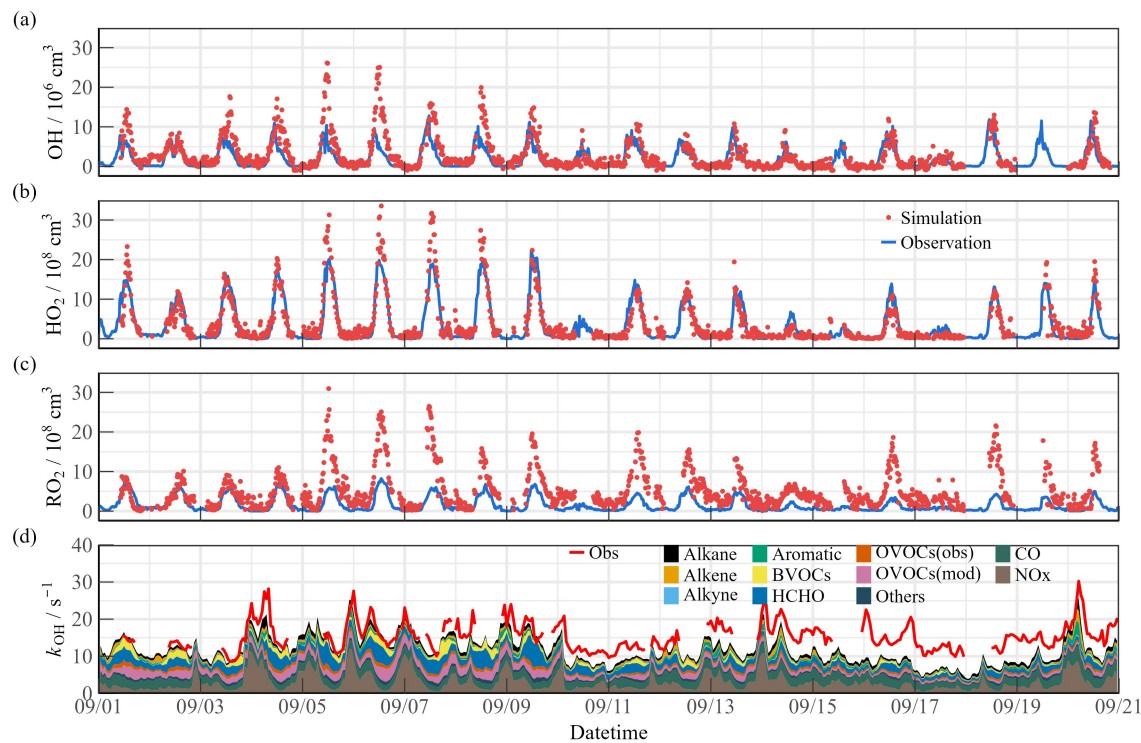

**Fig. 4.** Timeseries of the observed and modelled parameters for OH, HO₂ and $k_{OH}$ during the observation period.
**(a)** OH, **(b)** HO₂, **(c)** $k_{OH}$.

The significant variations in oxidation can be inferred from the disparities during different pollution periods (Fig. S6). During Semi I, there was a good agreement between the measurement and model for peroxy radicals during the daytime. The RACM2-LIM1 mechanism effectively replicated the morning OH radical concentration. However, following 10:00, NO gradually declined, and the increasing OH concentration could not be accounted for by the HO₂+NO formation channel, resulting in a maximum underestimation of $5.85 \times 10^6$ cm$^{-3}$ (Hofzumahaus et al., 2009; Lu et al., 2012). In the Semi II episode, OH was not underestimated in the low-NO regime, with a slight overestimation of HO₂ concentration. However, the simulated RO₂ concentration was only $3.78 \times 10^8$ cm$^{-3}$, whereas observations were 2.77 times larger than the simulation, indicating the existence of additional reaction pathways that likely propagated the OH→RO₂ conversion efficiency. A significant discrepancy of radicals existed in the heavy ozone concentration, with OH, HO₂, and RO₂ radicals concurrently underestimated at noon by $8.23 \times 10^6$ cm$^{-3}$, $3.94 \times 10^8$ cm$^{-3}$ and $11.59 \times 10^8$ cm$^{-3}$, respectively. The observed HO₂/RO₂ ratio approached 1:1, while the model reflected an unreasonable ratio of 3:1, indicating deficiencies in both primary sources and secondary propagation. The calculated reactivity seems to compare well with the observed OH reactivity at the start of the measurement period, but then there is evidence of missing OH reactivity after September 10th (Fig. 4(d)). Due to the limitations of available instruments, this observation only measured a limited number of OVOCs species, making it difficult to accurately quantify the contribution of larger aldehydes and ketones, carboxylic acids, nitrophenols, and other multifunctional species to $k_{OH}$ (Wang et al., 2024). Since the MCM mechanism considers more secondary formation reactions than the RACM2 mechanism, it can qualitatively assess the photochemical role of unmeasured OVOCs species in the atmosphere (Wang et al., 2022d). The additional modeled OVOCs by the MCM v3.3.1 mechanism contributed ~2.4 s$^{-1}$ to the missing OH reactivity (Fig. S7). During Heavy period, the reactivity of more model oxidation products increased the daytime $k_{OH}$ by about 5.1 s$^{-1}$. Therefore, the observed $k_{OH}$ can serve as an upper limit for sensitivity tests, thereby the full suite of radical measurement can be performed to explore the missing oxidation properties and ozone formation (Section 4.1).

387 Fig. 5 displays the diurnal profiles of the ROx budget during different episodes. In
388 Semi I, formaldehyde photolysis showed a higher contribution (38.6%), while HONO
389 photolysis (21.0%) and ozone photolysis (24.7%) accounted for similar proportions in
390 primary sources. The contribution of photolysis from other OVOCs was comparable to
391 that of ozonolysis reactions (7.2% vs. 4.8%). However, in Semi II, the decreased
392 oxidation level was attributed to lower ROx sources, despite the similar proportions.
393 During the Heavy period, the primary sources dramatically increased (up to ~10 ppb/h),
394 with HCHO photolysis contributing the most, alongside other sources at common levels
395 (ranging between 1.74 – 2.66 ppb/h) in the YRD region (Ma et al., 2022). Fast HCHO
396 oxidation dominated the radical primary source during heavy ozone pollution, which
397 contrasts with the dominant role of $HONO/O_3$ in other megacities (Yang et al., 2022; Tan
398 et al., 2017b; Yang et al., 2021a).

399 The radical removal rate during the daytime was generally balanced with production
400 contributions. In the morning, owing to high NOx concentrations, radical termination was
401 mainly dominated by $OH+NO_2$, $OH+NO$, $RO_2+NO$, and $RO_2+NO_2$. Furthermore, the
402 formation of peroxy nitrate accounted for a certain proportion (~5%). As NOx
403 concentrations decreased after 10:00, self-reactions in peroxy radicals became significant.

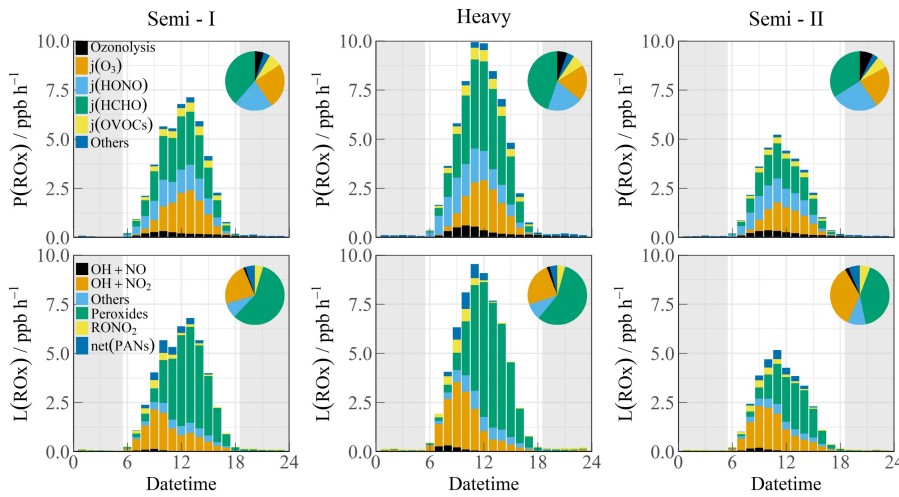

405 **Fig. 5.** The diurnal profiles of ROx budget during different polluted episodes (Semi I, Heavy, and Semi II). The
406 pie chart denotes proportions in different parts during the daytime (10:00-15:00). The grey areas denote nighttime.
407 By comparing the known sources and sinks for radicals, unknown processes for
408 initiation, transformation and termination can be determined in the experimental budget
409 analysis (Fig. S8). During the Semi I period, the production and destruction rates of $HO_2$,
410 $RO_2$, and total ROx radicals were very consistent, but a significant lack of a source term

for OH radicals was existed after 10:00. This missing source became more pronounced during the Heavy period, reaching 16 ppb/h at noon, which is close to the results observed by AIRPRO, but three times that observed by Heshan in PRD region(Tan et al., 2019b; Whalley et al., 2021). The ratio of OH production-to-destruction rate during the Semi II period was close to 1, indicating consistency between the observed results of OH, $HO_2$, $k_{OH}$, and other precursors(Whalley et al., 2018). However, the generation of $HO_2$ radicals in the morning was about twice as high as the removal rate, suggesting that there are contributions from unconsidered $HO_2$ radical removal channels (such as heterogeneous reactions)(Song et al., 2021). During the Heavy period, there was a rapid total removal rate of $RO_2$ radicals, reflecting the dominated $HO_2$ generation by the reaction of $RO_2$ radicals with NO. Although the $P(HO_2)$ and $D(HO_2)$ were quite in balance, the removal rate of $RO_2$ radicals far exceeded the known production rate (especially before 12:00). Previous work has shown that halogen chemistry (such as photolysis of nitryl chloride ($ClNO_2$)) could be an important source in the morning time, but this was not included in the calculation of ROx or $RO_2$ budget in this campaign(Tan et al., 2017b). The steady-state analysis for $HO_2$ radical in the London campaign emphasized that only by significantly reducing the observed $RO_2$-to-$HO_2$ propagation rate to just 15% could balance both $P(HO_2)$ and $D(HO_2)$, indicating that the $RO_2$-related mechanism for propagation to other radical species may not be fully understood(Whalley et al., 2018). Therefore, based on the current knowledge seems unlikely to explain the required source-sink difference of nearly 25 ppb/h in the $RO_2$ budget. Sensitivity analysis is needed to further infer the causes of the difference for the experimental budget analysis.

## 3.3 Oxidation comparison

The concentration of OH radicals during the daytime is a crucial indicator of atmospheric oxidation levels (Liu et al., 2021). Table 2 summarized radicals and related parameters for regions with similar latitudes ($32.0° \pm 2°$ N, $j(O^1D) \approx 2.5 \pm 0.5 \times 10^{-5}$ $s^{-1}$). The joint influence of solar radiation and local photochemistry resulted in megacities exhibiting intense oxidation levels in summer/autumn, characterized by OH radicals being maintained at approximately $10.0 \times 10^6$ $cm^{-3}$ at noon. Notably, an observation in Houston revealed an OH concentration of nearly $20.0 \times 10^6$ $cm^{-3}$, with $k_{OH}$ of 10 $s^{-1}$ (Mao

et al., 2010). In areas such as Los Angeles, Pasadena, and Tokyo, the propagation efficiency of radicals was restricted due to fresh anthropogenic emissions. OH concentrations were only half of those observed in other megacities, with higher inorganic-dominated $k_{OH}$ recorded (Pasadena, ~20 s$^{-1}$) (George et al., 1999; Griffith et al., 2016; Yugo Kanaya et al., 2007). In the TROPSTECT observation, the observed $k_{OH}$ exceeded the mean value at the same latitude (>15 s$^{-1}$). Additionally, during the Heavy episode, higher OH concentration (13.5×10$^6$ cm$^{-3}$) was found, comparable to the highest level at regions with similar latitude (Houston 2000/2006, (Mao et al., 2010)). Synchronous elevation in radical concentration and reactivity indicated a strong oxidation level in the YRD region.

The observations in the YRD region showed a stable conversion factor (OH-j(O$^1$D)) of 4±1 × 10$^{11}$ cm$^{-3}$ s, which was comparable to other megacities in the PRD, NCP, and SCB regions (Ma et al., 2022; Tan et al., 2019a). The corresponding slope between OH concentration and solar radiation was used to quantify the oxidation efficiency from photolysis, and it was observed that a higher slope of 5.3 × 10$^{11}$ cm$^{-3}$ s during the Heavy period indicated an active radical chemistry. This implies that there is a strong oxidation efficiency from photolysis in the YRD region.

During summer and autumn seasons, photochemical pollution is a common occurrence, as noted by (Tan et al., 2021). Analysis of radical concentration across different regions reveals that the YRD region exhibited concentrations higher than 10$^7$ cm$^{-3}$, slightly lower than in Guangzhou in 2006 but consistent with observations in other megacities (Ma et al., 2022; Tan et al., 2017a; Lu et al., 2012; Yang et al., 2021a). Conversely, winter is characterized by haze pollution (Ma et al., 2019). An urban site in Shanghai reported a peak OH concentration of 2.6×10$^6$ cm$^{-3}$, closely resembling the 1.7 – 3.1×10$^6$ cm$^{-3}$ range found in polluted winter atmospheres (Zhang et al., 2022a). Although no significant regional disparities in oxidation levels were detected in agglomerations, attention should be directed to the YRD region due to its elevated radical concentration, reactivity, and photolysis efficiency, signaling the need to investigate its role in radical chemistry.

**Table 2.** Summary of radical concentrations and related species concentrations at regions with similar latitude and megapolitan areas in China. All data are listed as the average in noontime (11:00~13:00).

| Location | Latitude | Year | OH (10$^6$ cm$^{-3}$) | $k_{OH}$ (s$^{-1}$) | j(O$^1$D) (10$^{-5}$ s$^{-1}$) | Slope (10$^{11}$ cm$^{-3}$ s) | X (ppb) | References |
|---|---|---|---|---|---|---|---|---|

| Regions with similar latitude | | | | | | | | |
|---|---|---|---|---|---|---|---|---|
| Los Angeles | 34.1° N | Sep 1993 | 6.0 | - | - | - | - | (George et al., 1999) |
| Nashville | 36.2° N | Jun–Jul 1999 | 10.0 | 10.2 | 3.0 | 3.3[c] | - | (Martinez et al., 2003) |
| Houston | 29.7° N | Aug 2000 | 20.0 | 9.0[b] | 3.0 | 6.7[c] | - | (Mao et al., 2010) |
| Tokyo | 35.6° N | Jul–Aug 2004 | 6.3 | - | 2.5 | 3.0 | - | (Yugo Kanaya et al., 2007) |
| Houston | 29.7° N | Sep 2006 | 15.0 | 11.0 | 3.1 | 5.0[c] | - | (Mao et al., 2010) |
| Pasadena | 34.1° N | May-Jun 2010 | 4.0 | 20.0 | 2.5 | 1.6[c] | - | (Griffith et al., 2016) |
| Taizhou | 32.6° N | May-Jun 2018 | 10.6 | 10.8[a] | 2.1 | 4.8 | 0.10 | (Ma et al., 2022) |
| Chengdu | 30.7° N | Aug 2019 | 10.0 | 8.0 | 2.2 | 4.1 | 0.25 | (Yang et al., 2021a) |
| TROPSTECT (Heavy) | 31.9° N | Sep 2020 | 13.5 | 16.0 | 2.6 | 5.3 | 0.50 | This work |
| TROPSTECT (Semi) | 31.9° N | Sep 2020 | 7.2 | 14.2 | 2.4 | 3.1 | 0.25 | This work |
| Regions in megapolitan areas in China | | | | | | | | |
| Guangzhou (PRD) | 23.5° N | Jul 2006 | 12.6 | 17.9 | 3.5[b] | 4.5 | 0.85 | (Lu et al., 2012) |
| Wangdu (NCP) | 38.7° N | Jun–Jul 2014 | 8.3 | 15.0 | 1.8 | 4.5 | 0.10 | (Tan et al., 2017b) |
| Beijing (NCP) | 39.9° N | May–Jun 2017 | 9.0 | 30.0 | 2.4 | 3.8[c] | ~0 | (Whalley et al., 2021) |
| Taizhou (YRD) | 32.6° N | May-Jun 2018 | 10.6 | 10.8[a] | 2.1 | 4.8 | 0.10 | (Ma et al., 2022) |
| Shenzhen (PRD) | 22.6° N | Sep-Oct 2018 | 4.5 | 21.0 | 1.8 | 2.4 | 0.10 | (Yang et al., 2022) |
| Chengdu (SCB) | 30.7° N | Aug 2019 | 9.0 | 8.0 | 2.2 | 4.0 | 0.25 | (Yang et al., 2021a) |
| Hefei (YRD) | 31.9° N | Sep 2020 | 10.4 | 14.3 | 2.4 | 4.4 | 0.30 | This work |

[a] The modeled $k_{OH}$.
[b] Value only in the afternoon.
[c] Using the ratio of OH / j(O¹D)

# 4 Discussion

## 4.1 Measurement–model reconciliation for radicals

### 4.1.1 OH underestimation

Full suite of OH, HO$_2$, RO$_2$ and $k_{OH}$ was utilized in the TROPSTECT campaign to untangle a thorough understanding of oxidation mechanisms where base model failed. One specific phenomenon was the absence of an OH source in situations where NO levels gradually decreased after 10:00. A sensitivity test was conducted introducing a species X, analogous to NO, to enhance OH regeneration (Fig. 6, RO$_2$→HO$_2$ and HO$_2$→OH) (Hofzumahaus et al., 2009). It was found that the addition of as little as 0.25 ppb X was sufficient to compensate for the full magnitude of the OH underestimation in the low NO region (Fig. 6). The employment of the X mechanism not only accelerated OH regeneration but also augmented the removal channel of peroxy radicals, which consequently led to a reduction in both HO$_2$ and RO$_2$ radical concentrations compared to the base scenario.

The underdetermined radical sources in China were corrseponding to the oxidation
level (Ma et al., 2022; Tan et al., 2017a; Lu et al., 2012; Yang et al., 2021a; Wang et al.,
2019b). The required X level typically ranged from 0.1 to 0.3 ppb, with the exception of
the Backgarden observation which required 0.85 ppb X, as indicated in Table 2 (Lu et al.,
2012). A minimum limit of 0.1 ppb X was established to account for any missing
reactivity (Ma et al., 2022). Notably, throughout the entire observation, a strong
agreement between the modeled and observed OH was achieved when a mixture of 0.25
ppb X was incorporated into the base scenario, consistent with the the order of magnitude
in Chengdu (Yang et al., 2021a). During the Heavy period, the augmented
photochemistry resulted in complex oxidation, necessitating an additional missing OH
source equivalent to 0.5 ppb X to fully address the underestimation of OH.

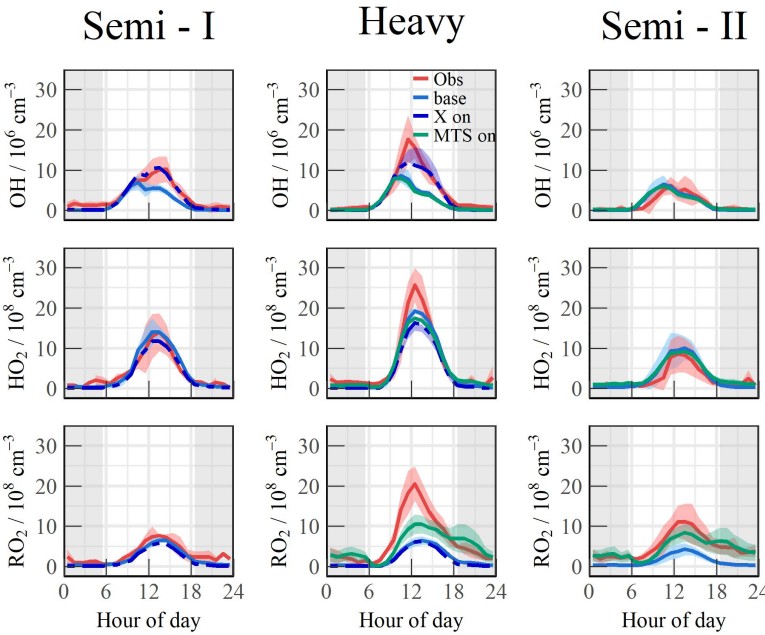


**Fig. 6.** The mean diurnal profiles of measured and modeled OH, $HO_2$ and $RO_2$ concentrations at different
scenarios. Sensitivity tests included three scenarios (Scenario 1: base case; Scenario 2: X mechanism on. The dashed
line represented the performance of 0.25 ppb X introduced in the Semi I and Heavy episodes, and the blue shadow
denoted the upper limit for X influence (0.5 ppb); Scenario 3: monoterpene mechanism on; Both API and LIM were
added into the base model as upper and lower limits for the influence of monoterpenes, and the mean of the two values
was taken as the average effect. The grey areas denote nighttime.
Missing OH sources are closely related to the chemistry of OVOCs(Yang et al.,
2024a; Qu et al., 2021). Reactive aldehyde chemistry, particularly the autoxidation of
carbonyl organic peroxy radicals ($R(CO)O_2$) derived from higher aldehydes, is a
significant OH regeneration mechanism that has been shown to contribute importantly to
OH sources in regions with abundant natural and anthropogenic emissions during warm

seasons(Yang et al., 2024b). In this study, the higher aldehyde mechanism (HAM) by Yang et al was parameterized into the base model to test new insights into the potential missing radical chemistry (Fig. 7). The results indicate that the contribution of the HAM mechanism to OH radicals in different episodes ranged between 4.4% - 6.0%, while the concentrations of $HO_2$ and $RO_2$ radicals increased by approximately 7.4% and 12.5%, respectively.

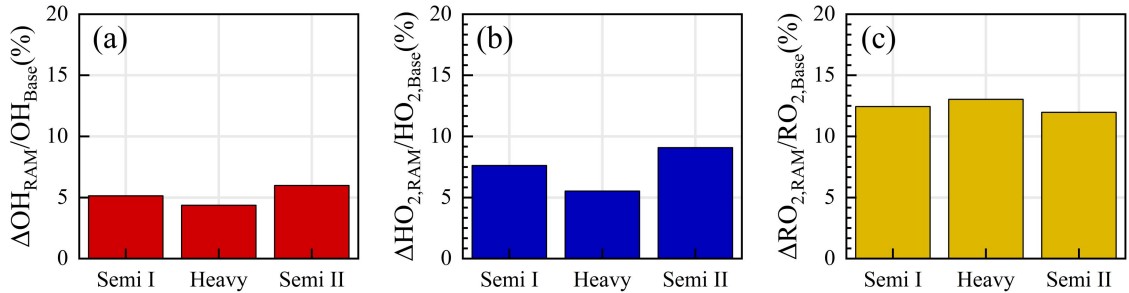

**Fig. 7.** The response of **(a)** OH, **(b)** $HO_2$ and **(c)** $RO_2$ radicals to the Higher Aldehyde Mechanism (HAM) in different episodes (Semi I, Heavy, and Semi II) in diurnal time (10:00-15:00).

### 4.1.2 $RO_2$ underestimation

The base scenario in Semi II is capable of accurately reproducing the concentrations of OH and $HO_2$ radicals within the data uncertainty. However, the simulated $RO_2$ concentration by the base model is only $3.78 \times 10^8$ $cm^{-3}$, which does not align with the observed oxidation levels in YRD, indicating a clear discrepancy. This underestimation is similarly evident in the APHH observation in Beijing, as the highest observed concentration of $RO_2$ radicals reached $5.5 \times 10^9$ $cm^{-3}$, far exceeding the level predicted by the MCM v3.3.1 mechanism (Whalley et al., 2021). The failure to reproduce the $RO_2$ concentration reflects the inadequacy of the mechanisms related to $RO_2$ radicals due to diverse oxidation reactions. This issue is further elucidated by previous studies, which highlighted the possibility of certain VOCs undergoing more intricate isomerization or fragmentation steps to sustain the long lifetime of $RO_2$ radicals (Whalley et al., 2018; Whalley et al., 2021).

The union of $k_{OH}$ and $RO_2$ measurement can help reveal the magnitude of missing $RO_2$ as a hypothesis of sensitivity analysis. An additional reaction was added to the base model in a previous research, converting OH into C96O2 (the oxidation product of α-pinene) with a reaction rate equal to the missing reactivity, to explore the source of the missing $RO_2$ radicals(Whalley et al., 2021). Discrepancy of OH reactivity (~3 – 5 $s^{-1}$)

between measurement and model suggested that an additional driving force was necessary to complete the OH to $RO_2$ step. In the TROPSPECT campaign, approximately 0.4 ppb of monoterpene was introduced into the base scenario as the chemical reactions of complex alkoxy radicals, which is similar to an atmospheric level in the EXPLORE-2018 campaign, the YRD region (Wang et al., 2022b). The RACM2 mechanism identified α-pinene (API) and limonene (LIM) as representative monoterpenes species. Sensitivity tests were conducted by incorporating API and LIM into the 'MTS on' and 'MTS+X on' scenarios, respectively (Ma et al., 2022). The mean of these values was considered the average effect of monoterpenes chemistry, and depicted as the green line in Fig. 6. In the 'MTS on' scenario, the chemistry of peroxy radicals in Semi II was reasonably described by introducing the source of complex alkoxy radicals, and the obs-to-mod ratio of peroxy radicals decreased from 2.2 to 1.3. Furthermore, the introduction of additional complex alkoxy radicals had minimal impact on HOx chemistry, with changes in daytime OH and $HO_2$ concentrations of less than $5\times10^5$ cm$^{-3}$ and $2.5\times10^7$ cm$^{-3}$, respectively. This demonstrates the robustness of HOx radical in response to potential monoterpene.

Higher aldehyde chemistry is a concrete manifestation of verifying the aforementioned hypothesis for $RO_2$ sources(Yang et al., 2024b). The autoxidation process of $R(CO)O_2$, encompasses a hydrogen migration process that transforms it into the ·OOR(CO)OOH radical(Wang et al., 2019b). This radical subsequently reacts with NO to yield the ·OR(CO)OOH radical. The ·OR(CO)OOH radical predominantly undergoes two successive rapid hydrogen migration reactionss, ultimately resulting in the formation of $HO_2$ radicals and hydroperoxy carbonyl (HPC). Consequently, the HAM mechanism extends the lifetime of the $RO_2$ radical, providing a valuable complement to the unaccounted sources of $RO_2$ radicals. As depicted in Fig. 7, the incorporation of the HAM mechanism results in an approximate 7.4% and 12.5% increase in the concentrations of $HO_2$ and $RO_2$ radicals, respectively. It is important to note that the total concentrations of primary emitted aldehydes and the HPC group may be underestimated, which could lead to the aforementioned analysis being conservative in nature. Further exploration of the unaccounted sources of $RO_2$ radicals will be presented in Section 4.3.

**4.1.3 P(Ox) underestimation**

Upon completing the hypothetical investigation into the radical underestimation, a

sensitivity comparison between observed and modeled P(Ox) was conducted across the
entire range of NO concentrations, as depicted in Fig. 8(a)(b). With increasing NO
concentration, the overall P(Ox) amplified, reaching a maximum of approximately 30
ppb/h. This variation has been validated through multiple observations in Wangdu, APHH,
and other studies (Tan et al., 2017b; Whalley et al., 2021; Whalley et al., 2018). However,
the imperfect understanding of the mechanisms related to peroxy radicals ultimately leads
to misjudgment of the ozone production process in high NO regimes, with a degree of
underestimation close to 10 times, as illustrated in Fig. 8(b).

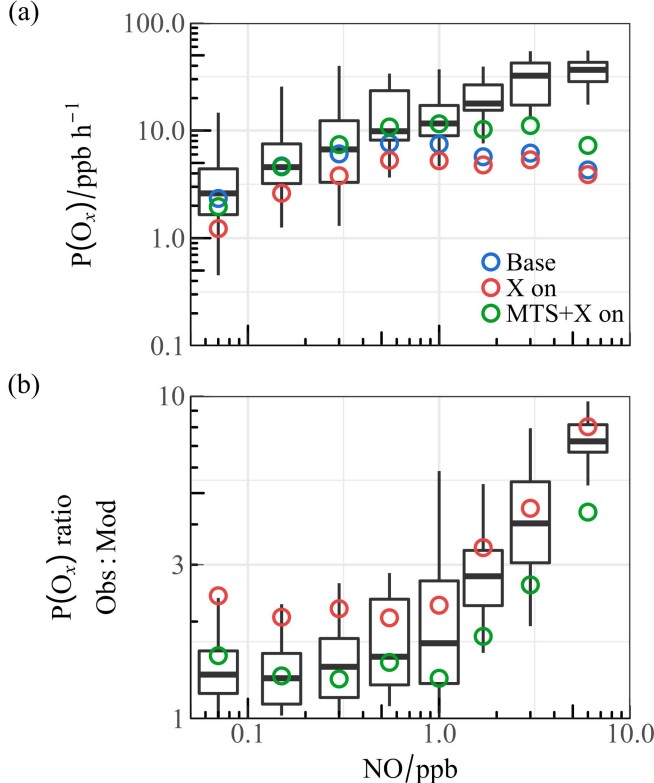


**Fig. 8.** The relationship between NO and **(a)** P(Ox), **(b)** P(Ox) (Obs:Mod). Boxplot diagrams are used to illustrate
the minimum, 25th percentile, median, 75th percentile, and maximum values of the observed dataset. The circles
represent the median values for the base model as well as for different mechanisms added to the model within various
ranges.

Although the inclusion of the X mechanism improves the agreement between

simulated and observed OH concentrations in the low-NO range, it has a negative effect
on the P(Ox) simulation. The introduction of a major source of $RO_2$ can help address the
underestimation problem in the base scenario, as the lack of $RO_2$ species and related
reaction rates is an important factor leading to deviations in the simulation of ozone

production rates (Tan et al., 2017a). The combination of the X mechanism and monoterpene chemistry is shown to better harmonize the relationship between $HO_2$ and $RO_2$. Notably, the deficiency in the ozone generation mechanism was adequately explained within a certain range in the 'MTS+X On' scenario, leading to an enhancement in the simulation performance of P(Ox) in the high NOx region (Fig. 8(b)). Therefore, reasonable simulation of the concentration of peroxy radicals is key to accurately quantifying the process of ozone generation.

## 4.2 Effect of mechanism reconciliation on oxidation

Both radical concentration and oxidation coordinating deficiency are worthy of examine (Fig. S9). To eliminate the influence of non-photolytic processes, only the daytime concentration range with $j(O^1D)$ greater than $5 \times 10^{-6}$ $s^{-1}$ was selected. The boxplots illustrate the ratio of observation to simulation (base model), with the circles representing the average values after integrating different mechanisms into the base scenario. In the low NO regime (NO < 1 ppb), the OH underestimation was consistently prominent as NO concentration decreased, and the base model was able to reasonably reflect the $HO_2$ distribution contrastly. As NO levels increased, the simulated OH concentration aligned well with the observation, but both $HO_2$ and $RO_2$ concentrations exhibited underprediction. $RO_2$ underestimation extended across the entire NO range, and could rise to over 10 times when NO levels reached about 10 ppb. Sensitivity tests based on the full suite of radical measurement revealed that the X mechanism accelerated OH regeneration, and the introduction of larger $RO_2$ alleviated the absence of certain sources by 2 to 4 times.

The coordinate ratios of radical serves as another test for ROx propagation (Fig. 9). The observed $HO_2$/OH ratio is approximately 100, declining to some extent as the concentration of NO increases, which is consistent with previous studies (Griffith et al., 2016; Griffith et al., 2013). However, the base model does not accurately replicate the curve depicting the change in $HO_2$/OH ratio, as shown in Fig. 9 (a). At low NO levels, the ratio significantly overestimated and shows a steeper decline compared to the base scenario as NO levels increase. Furthermore, the observed $RO_2$/OH ratios remain around 100, whereas the predicted values are significantly underestimated when NO exceeds 1 ppb (refer to Fig. 9(b)). In terms of the observed $HO_2$/$RO_2$ ratio, it maintains a relatively

constant trend within the range of 0.5 – 1.5, while the model overestimated by more than
twice, highlighting an inconsistency between the conversion of $RO_2 \rightarrow HO_2$. The
incorporation of the X mechanism has proven to be effective in a balanced $HO_2/OH$ ratio
as illustrated in Fig.9(a), but amplifing the termination pathway for $HO_2$ and $RO_2$, which
altered the coordination between $RO_2$ and OH across the entire NO range (Fig. 9(b)). The
connection between unconditional OH source and larger $RO_2$ isomerization in chemically
complex environments is key to fully understanding tropospheric chemistry, and a better
coordination of $HO_2/OH$, $RO_2/OH$, and $HO_2/RO_2$ ratios are established by incorporating
additional mechanisms.

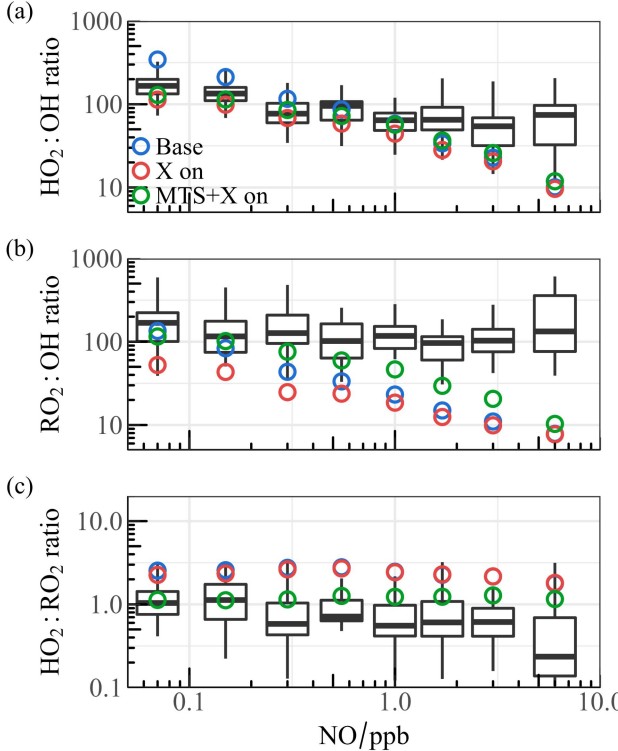

**Fig. 9.** The ratios for **(a)** $HO_2/OH$, **(b)** $RO_2/OH$, and **(c)** $HO_2/RO_2$ show a correlation with NO levels. Boxplot
diagrams are used to illustrate the minimum, 25th percentile, median, 75th percentile, and maximum values of the
observed dataset. The circles represent the median values for the base model as well as for different mechanisms added
to the model within various ranges.
The $HO_2/RO_2$ parameter was utilized to explore the transformation relationship
between $HO_2$ and $RO_2$ radicals. If $HO_2$ is formed from an $RO_2$ radical, it would result in
an $HO_2/RO_2$ radical concentration ratio of approximately 1. The $HO_2/RO_2$ ratios derived
from radical concentrations measured by laser-induced fluorescence instruments and
calculated using the MCM or RACM models were summarized in Fig. 10. In field studies,
the observed $HO_2/RO_2$ ratios were between 0.2 - 1.7 under low-NO conditions (NO < 1

ppb) and only 0.1 - 0.8 under high-NO conditions (3 < NO < 6 ppb). From the perspective of model-observation matching, except for three measurements in ClearfLo, ICOZA and AIRPRO-summer campaigns, the $HO_2/RO_2$ ratios in other regions could be reasonably reflected by the MCM or RACM2 mechanisms(Woodward-Massey et al., 2023; Whalley et al., 2021; Whalley et al., 2018; Färber et al., 2024). However, the ratio is generally underestimated under high NO conditions, reaching up to 5 times in ClearfLo. According to the latest chamber experiments, the $HO_2/RO_2$ radical concentration ratios for VOCs forming $HO_2$ are 0.6 for both one-step and two-step reactions. Therefore, the extremely low $HO_2/RO_2$ ratios observed in field campaigns can only be explained if almost all $RO_2$ radicals undergo multiple-step reactions before forming $HO_2$. During the TROPSTECT campaign, the observed $HO_2/RO_2$ remains at 1.1 and 0.8 under low-NO and high-NO conditions, respectively. After considering the complex sources of complex alkoxy radicals in the 'MTS+X' scenario, the simulated values of $HO_2/RO_2$ in both low-NO and high-NO regions match the observed values well.

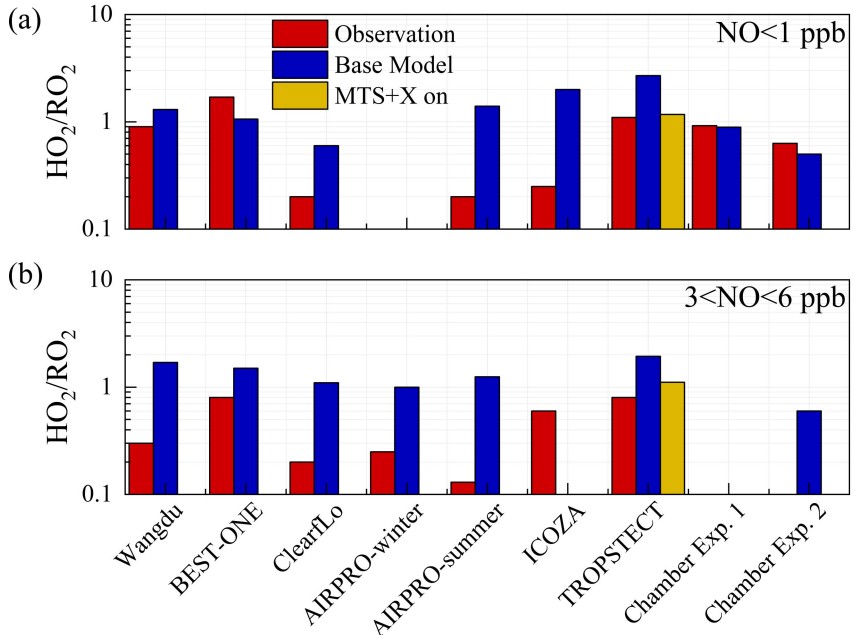

**Fig. 10.** Summary of the $HO_2/RO_2$ ratios derived from radical concentrations measured by laser-induced fluorescence instruments and calculated using the MCM or RACM models under **(a)** low-NO and **(b)** high-NO conditions. Charmber Exp. 1 and Charmber Exp. 2 denotes the parameters by single-step $HO_2$ formation and multi-step $HO_2$ formation determined in the chamber by (Färber et al., 2024).

## 4.3 Missing OVOCs sources influence ozone production

The consistency between model predictions and observed measurements for ozone

production, akin to the concentration ratio of $HO_2/RO_2$, is depicted in Fig. 11(a)(b). In
areas with low NO levels, the ratio of modeled to actual ozone production ranges from
0.5 to 2, with the exception of the ClearfLo and AIRPRO-summer
datasets(Woodward-Massey et al., 2023; Whalley et al., 2021). Conversely, under high
NO conditions (with NO concentrations between 3 and 6 ppbv), the ozone production
rate (P(Ox)) derived from measured radical concentrations typically exceeds that of the
base model's predictions by more than threefold. Laboratory experiments focusing on the
oxidation of representative VOCs suggest that ozone production can be enhanced by
approximately 25% for the anthropogenic VOCs under investigation(Färber et al., 2024).
The MTS+X scenario represents an effort to enhance the congruence between modeled
and measured radical concentrations. The incorporation of OVOCs and larger alkoxy
radicals derived from monoterpenes has refined the model-measurement agreement for
ozone formation under high NO conditions, reducing the discrepancy from 4.17 to 2.33.
This substantiates the hypothesis of sensitivity analysis concerning ozone generation, as
detailed in Section 4.2 and illustrated in Fig. S10.

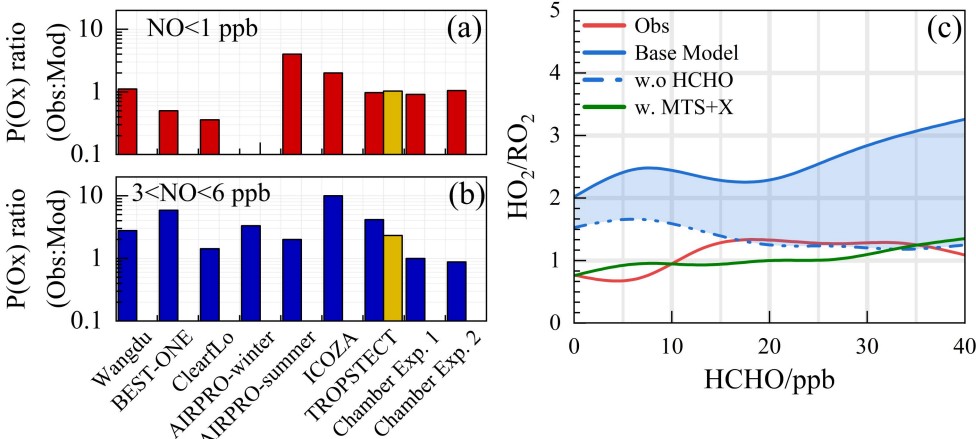


**Fig. 11.** Summary of the $P(Ox)_{Obs}/P(Ox)_{Mod}$ under **(a)** low-NO and **(b)** high-NO conditions.(c) The ratios for
$HO_2/RO_2$ show a correlation with HCHO levels. The blue shading represents the range of variation from constrained to
unconstrained formaldehyde conditions. Charmber Exp. 1 and Charmber Exp. 2 denotes the parameters by single-step
$HO_2$ formation and multi-step $HO_2$ formation determined in the chamber by (Färber et al., 2024).
The reasons for the discrepancy between simulated and observed values for ozone
production deserve further investigation. As depicted in Fig.11(c), the simulated
$HO_2/RO_2$ ratios display a robust positive correlation with photochemical activity,
fluctuating between 2 and 4. A notable feature during severe ozone pollution is the
intense distribution of formaldehyde, with an average concentration of $21.81 \pm 4.57$ ppb
(11:00 – 13:00). While formaldehyde acts as a precursor for $HO_2$ radicals, it does not
directly generate $RO_2$ radicals. The contributions of OVOCs to the ROx radical do not
exhibit the same intensity as formaldehyde, and the current mechanism encounters
difficulties in replicating formaldehyde concentrations (Fig. S11). The simulation of
formaldehyde concentrations using the MCM v3.3.1 mechanism has shown improvement,
indicating that the secondary formation of unmeasured species, such as OVOCs, will
feedback on $RO_2$ radical levels. When formaldehyde levels are unconstrained, the
simulated $HO_2/RO_2$ ratios align with observations, suggesting that under the prevailing
chemical mechanism, the photochemical efficiency of formaldehyde and other OVOCs is
similar. Therefore, an empirical hypothesis is proposed to amplify the concentration of
higher-order aldehydes by a factor of about 4, which is the proportion of formaldehyde
concentration underestimated by the model. The qualitative assessment of the impact of
missing aldehyde primary emissions on $RO_2$ radical concentrations was combined with
the HAM mechanism across the entire photochemical spectrum (Fig.S12). Enhanced
impact of aldehyde autoxidation in the presence of weak photochemical conditions could
alter the simulated levels of OH and $HO_2$ radicals by approximately 13.9% and 18.1%,
respectively. However, higher ALD concentrations will be achieved under intensive
photochemical conditions, leading to the gradual dominance of the sink channels for OH
+ OVOCs, with the effect of autoxidation mechanisms gradually decreasing. $RO_2$ radical
concentrations is notably more sensitive to the HAM mechanism, where incorporates
additional OVOCs, can enhance the simulation of $RO_2$ radical concentrations by 20 -
40%. Consequently, although limiting formaldehyde can partially offset the $HO_2$ radical
cycle and enhance the precision of HOx radical chemistry studies, additional
measurements should be undertaken for other OVOCs, coupled with the deployment of
full-chain radical detection systems, to accurately elucidate the oxidation processes under
severe ozone pollution conditions.

## 5 Conclusion

The full suite radical measurement of OH, $HO_2$, $RO_2$ and $k_{OH}$ was first deployed in
the YRD region (TROPSTECT) and encountered with a prolonged ozone pollution in
September 2020. The diurnal peaks of radicals exhibited considerable variation due to
environmental factors, showing ranges of 3.6 to $27.1\times10^6$ cm$^{-3}$ for OH, 2.1 to $33.2\times10^8$
cm$^{-3}$ for HO$_2$, and 4.9 to $30.5\times10^8$ cm$^{-3}$ for RO$_2$. Continuous $k_{OH}$ data fell within a range
of 8.6 – 30.2 s$^{-1}$, demonstrating the dominant behavior of organic species in diurnal
reactivity. Furthermore, observations in the YRD region were found to be similar to those
in other megacities, suggesting no significant regional differences in oxidation levels
were observed in agglomerations overall.
At a heavy ozone pollution episode, the oxidation level reached intensive compared
with other sites, and the simulated OH, HO$_2$, and RO$_2$ radicals provided by the
RACM2-LIM1 mechanism failed to adequately match the observed data both in radical
concentration and experimental radical budget. Sensitivity tests based on the full suite of
radical measurement revealed that the X mechanism accelerated OH regeneration, and
the introduction of larger alkoxy radicals alleviated the RO$_2$-related imbalance. The HAM
mechanism effectively complements the non-traditional regeneration of OH radicals,
improving by 4.4% - 6.0% compared to the base scenario, while the concentrations of
HO$_2$ and RO$_2$ radicals increased by approximately 7.4% and 12.5%, respectively. The
incorporation of complex processes enabled better coordination of HO$_2$/OH, RO$_2$/OH,
and HO$_2$/RO$_2$ ratios, and adequately addressed the deficiency in the ozone generation
mechanism within a certain range. Incorporation of OVOCs and larger alkoxy radicals
derived from monoterpenes improved the measurement-model consistency for ozone
formation under high NO conditions, reducing the discrepancy from 4.17 to 2.33, which
corroborates the hypothesis of sensitivity analysis in the context of ozone generation.
This study enabled a deeper understanding of the tropospheric radical chemistry at play.
Notably,
✓ A full suite of radical measurement can untangle the gap-bridge for the base model in
more chemically-complex environments as an hypothesis of sensitivity tests.
✓ Additional measurements targeting more OVOCs should also be conducted to fulfill
the RO$_2$-related imbalance, and then accurately elucidating the oxidation under
severe ozone pollution.

# Financial support

This work was supported by the National Key R&D Program of China (2022YFC3700301), the National Natural Science Foundation of China (62275250, U19A2044, 42030609), the Natural Science Foundation of Anhui Province (No. 2008085J20), the Anhui Provincial Key R&D Program (2022l07020022), and the Distinguished Program of Jianghuai Talents Program of Excellence (HYRCSTZ202401).

# Data availability

The data used in this study are available upon request (rzhu@aiofm.ac.cn).

# Author contributions

WQ Liu, PH Xie, RZ Hu contributed to the conception of this study. RZ Hu and GX Zhang performed the data analyses and manuscript writing. All authors contributed to measurements, discussed results, and commented on the paper.

# Competing interests

The contact author has declared that none of the authors has any competing interests.

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
