# Peer review of "Accurate Elucidation of Oxidation Under Heavy Ozone"

_EGUsphere, 2024_

## Author Comment (AC1)

Dear Editor Eleanor Browne and Referee,

Thanks for your suggestions which significantly help us to improve the manuscript. Hereby, we submit our responses and the manuscript has been revised accordingly. If there are any further questions or comments, please let us know.

Best regards

Guoxian Zhang on behalf of all co-authors

Key Lab. of Environmental Optics & Technology, Anhui Institute of Optics and Fine Mechanics, Chinese Academy of Sciences

230031 Hefei China

E-mail: gxzhang@aiofm.ac.cn

**Major Comments**

*1. Line 180: LIF groups now routinely use inlet-pre-injectors to chemically remove ambient OH prior to sampling (to determine their background signal for subtraction) to ensure an interference-free OH measurement. Wavelength modulation does not allow distinction between ambient OH and any OH internally generated within the reaction cell. A previous comparison exercise with a second LIF instrument at a different location does not ensure that the instrument (and the OH measurement presented here) is free from interferences. This needs to be acknowledged when discussing the model measurement comparison.*

**Reply:**

Thanks for your suggestion. During the TROPSTECT-YRD campaign, we did not use an inlet-pre-injector to determine the chemical background of OH radical. We acknowledge your point that the comparison exercise with a second LIF instrument at a different location does not ensure that the instrument (and the OH measurement presented here) is free from interferences. We will discuss whether internal interference exists in AIOFM-LIF from the following aspects:

First of all, literature research shows that measurement interference is more related to the length of the inlet in the low-pressure cell (Griffith et al., 2016). In terms of system design, the AIOFM-LIF system uses a short-length inlet design to minimize this and other unknown disturbances (the distance from radical sampling to flourescence excitation is ~150 mm).

Additionally, potential interference may exist when the atmosphere contains abundant alkenes, ozone, and BVOCs, indicating that environmental conditions play leading roles in OH interferences (Mao et al., 2010; Fuchs et al., 2016; Novelli et al., 2014). In the previous comparison exercise with a LIF instrument deployed an inlet pre-injector (PKU-LIF), the ozonolysis interference on the measurement consistency of both systems was excluded under high-VOCs conditions (Zhang et al., 2022).

We have compared the chemical conditions during the intercomparison

experiment and the current environmental conditions. Overall, the key parameters related to ozonolysis reactions ($O_3$、alkenes、isoprene and NOx) in TROPSTECT-YRD were similar to those during the comparison experiment, which is not conducive to generating potential OH interference.

**Table.** Comparison of key parameters related to ozonolysis reactions ($O_3$、 alkenes、 isoprene and NOx) between TROPSTECT-YRD and the intercomparison experiment. All the values are the diurnal average (10:00-15:00).

| Species | Intercomparison | TROPSTECT-YRD |
| --- | --- | --- |
| $O_3$ (ppb) | 71.02 | 76.25 |
| Alkenes (ppb) | 1.29 | 0.67 |
| Isoprene (ppb) | 0.67 | 0.86 |
| NOx (ppb) | 5.65 | 6.55 |

To provide direct evidence on the OH chemical background signal, we conducted another atmospheric oxidation observation in the same location (Science Island background station  in Hefei) and season (September, Autumn in 2022) in 2022, using chemical modulation methods to measure the chemical background of OH radicals in AIOFM-LIF instrument. The environmental conditions during ozone pollution (2022.9.29-2022.10.3) are shown in the Fig. S3, with daytime peaks of ozone concentration above 75 ppb, accompanied by alkene species approaching ~10 ppb. The diurnal concentration of isoprene was also a high level (>1 ppb). The chemical conditions are more favourable to induce OH interference than the TROPSTECT-YRD site. However, the OH concentrations achieved by chemical modulation ($OH_{chem}$) and wavelength modulation ($OH_{wav}$) were in good agreement. No obvious chemical background was observed by deploying an inlet pre-injector. Therefore, it is not expected that OH measurement in the present study was affected by internal interference.

[Figure]

**Fig. S3.** Results of an additional atmospheric oxidation observation experiment in the same location and season in 2022. **(a)** Ozone concentration **(b)** Concentrations of alkene and isoprene, respectively. **(c)** The OH concentrations achieved by chemical modulation (OH$_{chem}$) and wavelength modulation (OH$_{wav}$).

We added the detailed description in Line 187-197.

**Revision:**

Line 187-197: An additional atmospheric oxidation observation was conducted in the same location and season in 2022 with a chemical modulation method to determine the chemical background of OH radicals (Fig. S3). During the ozone pollution (2022.9.29-2022.10.3), the daytime peaks of ozone concentration above 75 ppb, accompanied by alkene species approaching ~10 ppb. The diurnal concentration of isoprene was also a high level (>1 ppb). The chemical conditions are more favourable to induce OH interference than in the TROPSTECT campaign, while the OH concentrations achieved by chemical modulation (OH$_{chem}$) and wavelength modulation (OH$_{wav}$) were in good agreement. No obvious chemical background was observed by deploying an inlet pre-injector. Therefore, it is not expected that OH measurement in the present study was affected by internal interference.

2. *Section 2.2.2: the description of the OH reactivity instrument lacks adequate detail. How is OH generated? Via the photolysis of ambient or generated ozone?*

*What was the initial OH concentration generated? Flow rate and pressure in the flow-tube?*

**Reply:**

Thank you for your reply. OH radicals are generated by laser photolysis of ambient ozone, using a laser pulse with a wavelength of 266 nm. Under conditions of 80 ppb $O_3$ and 8000 ppm water vapor concentration, the concentration of OH radicals produced in the flow tube remains at the order of $10^9$ cm$^{-3}$. The flow tube is at ambient pressure, with a gas flow rate of 17 SLM. We have supplemented the detailed description for the OH reactivity measurement instrument in Line 219-230.

**Revision:**

Line 219-230: The configuration structure for $k_{OH}$ measurement has been detailed in a previous study(Liu et al., 2019). The flow tube in the OH production-reaction unit is at ambient pressure, with a gas flow rate of 17 SLM. A pulsed laser beam (266 nm with an average power of 15 mJ) is output from a frequency-quadrupled Nd:YAG laser, which generates stable OH radical through flash photolysis of ambient ozone in the flow tube. Consistent and stable production of OH radicals is ensured by maintaining a stable concentration of reactants, flow field, and laser energy. Under conditions of 80 ppb $O_3$ and 8000 ppm water vapor concentration, OH radicals produced in the flow tube remains at the concentration order of $10^9$ cm$^{-3}$. Subsequently, the OH radicals are sampled through a nozzle into a fluorescence cell. The OH fluorescence signal is then detected using laser pump and probe techniques and is fitted to calculate the slope of OH decay ($k_{OH}$). The detection accuracy, achieved with an integration time of 180 s, is 0.3 s$^{-1}$ (1$\sigma$).

3. *Section 2.3: A comprehensive list of model constraints should be provided. Which NMHCs were measured?*

**Reply:**

Thank you for your reply. The comprehensive list of model constraints was provided in Table S3. The measured NMHCs include 29 alkanes, 11 alkenes, 15

aromatics, as well as acetylene and isoprene, and the specific names are also listed in Table S3. We have supplemented the detailed description in Line 237-239.

**Revision:**

Line 237-239: The comprehensive list of model constraints was provided in Table S3. The measured NMHCs include 29 alkanes, 11 alkenes, 15 aromatics, as well as acetylene and isoprene.

**Table.S3.** The comprehensive list of model constraints.

| Categories | Species |
|---|---|
| Meteorology | Temperature, Relative humidity, Pressure, Jvalues |
| Trace gases | $O_3$, NO, $NO_2$, $SO_2$, CO, PAN, HONO |
| Alkanes | methane, ethane, propane, n-butane, isobutane, cyclopentane, n-pentane, isopentane, cyclohexane, methyl cyclopentane, 2,3-dimethyl butane, 2,2-dimethyl butane, n-hexane, 2-methyl pentane, 3-methyl pentane, methyl cyclohexane, n-heptane, 2-methyl hexane, 2,3-dimethyl pentane, 2,4-dimethyl pentane, 3-methyl hexane, n-octane, 2,3,4-trimethyl pentane, 2-methyl heptane, 3-methyl heptane, 2,2,4-trimethyl pentane, n-nonane, n-decane, n-undecane, n-dodecane |
| Alkenes | ethene, propene, 1,3-butadiene, 1-butene, cis-2-butene, trans-2-butene, 1-pentene, cis-2-pentene, trans-2-pentene, 1-hexene, styrene |
| BVOCs | isoprene |
| Alkynes | acetylene |
| Aromatics | benzene, toluene,ethyl benzene, o-xylene, m-xylene, n-propyl benzene, isopropyl benzene, p-ethyl toluene, o-ethyl toluene, m-ethyl toluene, 1,2,4-trimethyl benzene, 1,3,5-trimethyl benzene, 1,2,3-trimethyl benzene, p-diethyl benzene, m-diethyl benzene |
| OVOCs | HCHO, acetaldehyde, MACR, MVK |

**4. Line 223: Was the model unconstrained to O3 and NO2 in this scenario?**

**Reply:**

Thank you for your reply. In the base scenario, the species involved in Table S3 are constrained as boundary conditions. In the ozone-simulation mode that mentioned in Fig.S4, the model unconstrained to $O_3$ and NO on the basis of the base scenario. We summarized the sensitive test scenarios used in the manuscript in Table 1, and the detailed description in Line 237-239&241-247.

**Table.1.** The sensitive test scenarios utilized to improve the model-measurement consistency between OH, $HO_2$ and $RO_2$ radicals.

| Scenario | Configuration | Purpose |
|---|---|---|

| | | |
|---|---|---|
| Base | RACM2 updated with isoprene reaction scheme (LIM) | The base case with the species involved in Table S3 are constrained as boundary conditions. |
| X on | As the base scenario, but add the X mechanism, and the X level is between 0.25 - 0.5 ppb. | To untangle the missing OH source where base scenario failed. |
| MTS on | As the base scenario, but add a monoterpene source, and the monoterpene level is ~0.4 ppb. | Utilizing monoterpene-derived $RO_2$ to represent the alkoxy radicals with rather complex chemical structures. |
| MTS+X on | As the base scenario, but both the X mechanism and monoterpene source are considered. | To consider both the missing OH and $RO_2$ sources. |
| HAM on | As the base scenario, but add the reactive aldehyde chemistry. | To provide a test of whether the proposed mechanism can explain the missing OH source. |
| HAM on (4 × ALD) | As the base scenario, but add the reactive aldehyde chemistry, and the concentration of ALD was amplified by a factor of 4. | To quantify the impact of missing aldehyde primary emissions on ROx chemistry. |
| Ozone simulation | As the base scenario, but remove the constraints of the observed ozone and NO concentrations. | To test the suitable lifetime for the base model. |
| HCHO simulation | As the base scenario, but remove the constraint of the observed HCHO concentration. | To test the simulation effect of the existing mechanism on formaldehyde concentration. |

**Revision:**

Line 237-239: The comprehensive list of model constraints was provided in Table S3. The measured NMHCs include 29 alkanes, 11 alkenes, 15 aromatics, as well as acetylene and isoprene.

Line 241-247: An ozone-simulation test was conducted to determine the suitable atmospheric lifetime ($\tau_D$) for the base model. At the lifetime of 24 hours, with a corresponding first-order loss rate of 1.1 cm/s (assuming a boundary layer height of 1 km), the simulated ozone concentration closely matched the observed values (Fig. S4). To improve the model-measurement consistency between OH, $HO_2$ and $RO_2$ radicals, a series of sensitivity analyses were performed to evaluate the impacts of potential mechanisms, as detailed in Table 1.

5. *Fig. 4 highlights that OVOCs contribute significantly to OH reactivity. Given that one of the major conclusions of the manuscript is that future measurement*

*campaigns should target more OVOCs, the individual OVOCs that are considered in this class should be provided. It would be beneficial to list all the VOCs that have been considered in all the different groups in a table. The calculated reactivity seems to compare well with the observed OH reactivity at the start of the measurement period, but then there is evidence of missing OH reactivity after the 10th, why is this? Was the contribution model-generated intermediates make to the calculated OH reactivity considered?*

**Reply:**

Thank you for your reply. We have listed the VOCs involved in the model simulation in Table S3 and have specifically detailed the contribution of OVOCs to OH reactivity (Fig. 4).

Table.S3. The comprehensive list of model constraints.

| Categories | Species |
|---|---|
| Meteorology | Temperature, Relative humidity, Pressure, Jvalues |
| Trace gases | $O_3$, NO, $NO_2$, $SO_2$, CO, PAN, HONO |
| Alkanes | methane, ethane, propane, n-butane, isobutane, cyclopentane, n-pentane, isopentane, cyclohexane, methyl cyclopentane, 2,3-dimethyl butane, 2,2-dimethyl butane, n-hexane, 2-methyl pentane, 3-methyl pentane, methyl cyclohexane, n-heptane, 2-methyl hexane, 2,3-dimethyl pentane, 2,4-dimethyl pentane, 3-methyl hexane, n-octane, 2,3,4-trimethyl pentane, 2-methyl heptane, 3-methyl heptane, 2,2,4-trimethyl pentane, n-nonane, n-decane, n-undecane, n-dodecane |
| Alkenes | ethene, propene, 1,3-butadiene, 1-butene, cis-2-butene, trans-2-butene, 1-pentene, cis-2-pentene, trans-2-pentene, 1-hexene, styrene |
| BVOCs | isoprene |
| Alkynes | acetylene |
| Aromatics | benzene, toluene,ethyl benzene, o-xylene, m-xylene, n-propyl benzene, isopropyl benzene, p-ethyl toluene, o-ethyl toluene, m-ethyl toluene, 1,2,4-trimethyl benzene, 1,3,5-trimethyl benzene, 1,2,3-trimethyl benzene, p-diethyl benzene, m-diethyl benzene |
| OVOCs | HCHO, acetaldehyde, MACR, MVK |

In Fig. 4, $k_{OVOCs}$ are categorized into three groups: $k_{OVOCs(Obs)}$, $k_{OVOCs(Model)}$, and $k_{HCHO}$. Given the significance of formaldehyde photolysis, the contribution of HCHO to $k_{OVOCs}$ is distinguished. $k_{OVOCs(Obs)}$ encompasses species observed in addition to formaldehyde, such as acetaldehyde (ACD) and the oxidation products of isoprene

(MACR and MVK). Intermediates generated by the model, including glyoxal (GLY), methylglyoxal (MGLY), higher aldehydes (ALD), ketones (KET), methyl ethyl ketone (MEK), and methanol (MOH), are classified as $k_{OVOCs(Model)}$. Upon considering $k_{OVOCs(Model)}$, the calculated reactivity seems to compare well with the observed OH reactivity at the start of the measurement period, but then there is evidence of missing OH reactivity after September 10th (Fig.4(d)).

[Figure]

**Fig. 4.** Timeseries of the observed and modelled parameters for OH, HO$_2$ and $k_{OH}$ during the observation period. **(a)** OH, **(b)** HO$_2$, **(c)** $k_{OH}$.

Due to the limitations of available instruments, this observation only measured a limited number of OVOCs species, making it difficult to accurately quantify the contribution of larger aldehydes and ketones, carboxylic acids, nitrophenols, and other multifunctional species to $k_{OH}$ (Wang et al., 2024). Since the MCM mechanism considers more secondary formation reactions than the RACM2 mechanism, it can qualitatively assess the photochemical role of unmeasured OVOCs species in the atmosphere (Wang et al., 2022b). The additional modeled OVOCs by the MCM v3.3.1 mechanism contributed ~2.4 s$^{-1}$ to the missing OH reactivity (Fig.S7). During Heavy period, the reactivity of more model oxidation products increased the daytime $k_{OH}$ by about 5.1 s$^{-1}$. Therefore, the observed $k_{OH}$ can serve as an upper limit for

sensitivity tests, thereby the full suite of radical measurement can be performed to explore the missing oxidation properties and ozone formation (Section 4.1).

[Figure]

**Fig. S7.** Timeseries of the observed and modelled $k_{OH}$ during the observation period.

We added the detailed description in Line 345-353&372-386.

**Revision:**

Line 345-353: $k_{OVOCs}$ are categorized into three groups: $k_{OVOCs(Obs)}$, $k_{OVOCs(Model)}$, and $k_{HCHO}$. Given the significance of formaldehyde photolysis, the contribution of HCHO to $k_{OVOCs}$ is distinguished. $k_{OVOCs(Obs)}$ encompasses species observed in addition to formaldehyde, such as acetaldehyde (ACD) and the oxidation products of isoprene (MACR and MVK). Intermediates generated by the model, including glyoxal (GLY), methylglyoxal (MGLY), higher aldehydes (ALD), ketones (KET), methyl ethyl ketone (MEK), and methanol (MOH), are classified as $k_{OVOCs(Model)}$. Upon considering $k_{OVOCs(Model)}$, the reactivity calculated prior to September 10th aligns quite well with the observed OH reactivity.

Line 372-386: The calculated reactivity seems to compare well with the observed OH reactivity at the start of the measurement period, but then there is evidence of missing OH reactivity after September 10th (Fig. 4(d)). Due to the limitations of available instruments, this observation only measured a limited number of OVOCs species, making it difficult to accurately quantify the contribution of larger aldehydes and ketones, carboxylic acids, nitrophenols, and other multifunctional species to $k_{OH}$ (Wang et al., 2024). Since the MCM mechanism considers more secondary formation reactions than the RACM2 mechanism, it can qualitatively assess the photochemical role of unmeasured OVOCs species in the atmosphere (Wang et al., 2022b). The additional modeled OVOCs by the MCM v3.3.1 mechanism contributed ~2.4 s⁻¹ to

the missing OH reactivity (Fig. S7). During Heavy period, the reactivity of more model oxidation products increased the daytime $k_{OH}$ by about 5.1 s$^{-1}$. Therefore, the observed $k_{OH}$ can serve as an upper limit for sensitivity tests, thereby the full suite of radical measurement can be performed to explore the missing oxidation properties and ozone formation (Section 4.1).

**6. *Line 444-455: This section discusses the inclusion of monoterpenes in the model. The authors need to describe how RACM2 treats the oxidation of alpha-pinene and how this compares to the MCM mechanism for alpha pinene.***

**Reply:**

Thank you for your reply. The oxidation processes of α-pinene and limonene related to RACM2 mechanism have been listed in Table S4, including the oxidation reactions with OH/O$_3$/NO$_3$, as well as the reactions of the derived alkoxy radicals (APIP) with NO/NO$_3$/HO$_2$ and the self-reactions among peroxy radicals. In the MCM mechanism, the derivative of α-Pinene, C96O2, could undergo four RO$_2$→RO$_2$ propagations before returning to HO$_2$ radicals. An additional reaction was added to the base model in a previous research, converting OH into C96O2 (the oxidation product of α-pinene) with a reaction rate equal to the missing reactivity, to explore the source of the missing RO$_2$ radicals(Whalley et al., 2021). In the RACM2 mechanism, the peroxy radicals generated from α-pinene oxidation are classified as APIP and return to HO$_2$ radicals through subsequent reactions with NO. Therefore, we place greater emphasis on utilizing monoterpene-derived RO$_2$ in sensitive experiments to represent those RO$_2$ radicals with relatively complex chemical structures (Table 1).

**Table.S4.** Gas-phase kinetics for the monoterpene species in RACM2 mechanism. API and LIM stand for α-pinene and limonene, respectively; APIP and LIMP represents peroxy radicals derived from API and LIM, respectively; ETHP refers to peroxy radicals generated from ethane; KETP denotes peroxy radicals formed from ketones; ALD signifies C$_3$ and higher aldehydes; KET indicates ketones; OLNN pertains to the NO$_3$-alkene adduct that reacts to form carbonitrates and HO$_2$; OLND pertains to the NO$_3$-alkene adduct that reacts through decomposition; ACT signifies acetone; ORA1 denotes formic acid; ONIT represents organic nitrate; OP2 denotes higher organic peroxides; MO2 signifies methyl peroxy radical; MOH indicates methanol; ROH denotes C3 and higher alcohols; ACO3 represents acetyl peroxy radicals; ORA2 denotes acetic acid and other higher acids.

| Reaction | Reaction rate constant (cm$^3$s$^{-1}$) |
|---|---|
| API + OH → APIP | $1.21 \times 10^{-11}$exp(440/T) |
| API + O$_3$ → 0.85 × OH + 0.1 × HO$_2$ + 0.2 × ETHP + 0.42 × KETP + | $5.0 \times 10^{-16}$exp(-530/T) |

| Reaction | Rate |
|---|---|
| $0.14 \times CO + 0.02 \times H_2O_2 + 0.65 \times ALD + 0.53 \times KET$ | |
| $API + NO_3 \rightarrow 0.1 \times OLNN + 0.9 \times OLND$ | $1.19 \times 10^{-12}exp(490/T)$ |
| $APIP + NO \rightarrow 0.82 \times HO_2 + 0.82 \times NO_2 + 0.23 \times HCHO + 0.43 \times ALD + 0.44 \times KET + 0.07 \times ORA1 + 0.18 \times ONIT$ | $4.0 \times 10^{-12}$ |
| $APIP + HO_2 \rightarrow OP2$ | $1.5 \times 10^{-11}$ |
| $APIP + MO_2 \rightarrow HO_2 + 0.75 \times HCHO + 0.75 \times ALD + 0.75 \times KET + 0.25 \times MOH + 0.25 \times ROH$ | $3.56 \times 10^{-14}exp(708/T)$ |
| $APIP + ACO_3 \rightarrow 0.5 \times HO_2 + 0.5 \times MO_2 + ALD + KET + ORA2$ | $7.4 \times 10^{-13}exp(765/T)$ |
| $APIP + NO_3 \rightarrow HO_2 + NO_2 + ALD + KET$ | $1.2 \times 10^{-12}$ |
| $LIM + OH \text{-->} LIMP$ | $4.2 \times 10^{-11}exp(401/T)$ |
| $LIM + O_3 \text{-->} 0.85 \times HO + 0.1 \times HO_2 + 0.16 \times ETHP + 0.42 \times KETP + 0.02 \times H_2O_2 + 0.14 \times CO + 0.46 \times OLT + 0.04 \times HCHO + 0.79 \times MACR + 0.01 \times ORA1 + 0.07 \times ORA2$ | $2.95 \times 10^{-15}exp(-783/T)$ |
| $LIM + NO_3 \text{-->} 0.71 \times OLNN + 0.29 \times OLND$ | $1.22 \times 10^{-11}$ |
| $LIMP + NO \text{-->} HO_2 + NO_2 + 0.05 \times OLI + 0.43 \times HCHO + 0.68 \times UALD + 0.07 \times ORA1$ | $4.0 \times 10^{-12}$ |
| $LIMP + HO_2 \text{-->} OP_2$ | $1.5 \times 10^{-11}$ |
| $LIMP + MO_2 \text{-->} HO_2 + 0.192 \times OLI + 1.04 \times HCHO + 0.308 \times MACR + 0.25 \times MOH + 0.25 \times ROH$ | $3.56 \times 10^{-14}exp(708/T)$ |
| $LIMP + ACO_3 \text{-->} 0.5 \times HO_2 + 0.5 \times MO_2 + 0.192 \times OLI + 0.385 \times HCHO + 0.308 \times MACR + 0.5 \times ORA2$ | $7.4 \times 10^{-13}exp(765/T)$ |
| $LIMP + NO_3 \text{-->} HO_2 + NO_2 + 0.385 \times OLI + 0.385 \times HCHO + 0.615 \times MACR$ | $1.2 \times 10^{-12}$ |

Discrepancy of OH reactivity ($\sim 3 - 5$ $s^{-1}$) between measurement and model suggested that an additional driving force was necessary to complete the OH to $RO_2$ step. In the TROPSPECT campaign, approximately 0.4 ppb of monoterpene was introduced into the base scenario as the chemical reactions of complex alkoxy radicals, which is similar to an atmospheric level in the EXPLORE-2018 campaign, the YRD region (Wang et al., 2022a). Sensitivity tests were conducted by incorporating API and LIM into the 'MTS on' and 'MTS+X on' scenarios as the chemical reactions of complex alkoxy radicals, respectively (Ma et al., 2022). We added the detailed description in Line 535-555.

**Revision:**

Line 535-555: The union of $k_{OH}$ and $RO_2$ measurement can help reveal the magnitude of missing $RO_2$ as a hypothesis of sensitivity analysis. An additional reaction was added to the base model in a previous research, converting OH into C96O2 (the oxidation product of α-pinene) with a reaction rate equal to the missing reactivity, to explore the source of the missing $RO_2$ radicals(Whalley et al., 2021). Discrepancy of OH reactivity ($\sim 3 - 5$ $s^{-1}$) between measurement and model suggested that an

additional driving force was necessary to complete the OH to $RO_2$ step. In the TROPSPECT campaign, approximately 0.4 ppb of monoterpene was introduced into the base scenario as the chemical reactions of complex alkoxy radicals, which is similar to an atmospheric level in the EXPLORE-2018 campaign, the YRD region (Wang et al., 2022a). The RACM2 mechanism identified α-pinene (API) and limonene (LIM) as representative monoterpenes species. Sensitivity tests were conducted by incorporating API and LIM into the 'MTS on' and 'MTS+X on' scenarios, respectively (Ma et al., 2022). The mean of these values was considered the average effect of monoterpenes chemistry, and depicted as the green line in Fig. 6. In the 'MTS on' scenario, the chemistry of peroxy radicals in Semi II was reasonably described by introducing the source of complex alkoxy radicals, and the obs-to-mod ratio of peroxy radicals decreased from 2.2 to 1.3. Furthermore, the introduction of additional complex alkoxy radicals had minimal impact on HOx chemistry, with changes in daytime OH and $HO_2$ concentrations of less than $5\times10^5$ cm$^{-3}$ and $2.5\times10^7$ cm$^{-3}$, respectively. This demonstrates the robustness of HOx radical in response to potential monoterpene.

7. *Section 4.3: I found this section particularly difficult to follow. What do the authors mean by 'Special HCHO'? Could the authors provide the model predicted HCHO concentration (when left unconstrained to HCHO) relative to the HCHO concentration measured? The main conclusion of this section seems to be that other OVOC (that can act as a source of RO2) should be measured, but there is no discussion on what OVOCs were measured beyond HCHO; this detail needs to be included.*

**Reply:**

Thank you for your response. The term "special HCHO" mentioned in the manuscript aims to emphasize a phenomenon where formaldehyde has a high concentration distribution (with an average concentration of $21.81 \pm 4.57$ ppb at noon), but the contributions of OVOCs to the ROx radical do not exhibit the same intensity as formaldehyde, and the current mechanism encounters difficulties in replicating

formaldehyde concentrations. We acknowledge your point that this part of the description is too confusing. Therefore, we have changed the title of the relevant section to "4.3 Missing OVOCs sources influence ozone production" and adjusted the content of that section. We have removed the impact of formaldehyde on the length of the reaction chain and its oxidizing effect, focusing more on the diagnostic of the $HO_2/RO_2$ ratio on ozone formation to improve the readability of the manuscript.

The information on the measured OVOCs species has been integrated into Supplementary Table S3, and their contributions to $k_{OH}$ are discussed in detail on Lines 372-386 and in Supplementary Fig.S7. The comparison results between the simulated and measured values of formaldehyde concentrations are shown in the Fig. S11. The deposition time is set to 24 hours, and a comparative analysis has been conducted based on the MCM v3.3.1 mechanism and RACM2-LIM1 mechanism. The results show that the simulated formaldehyde concentrations are significantly lower than the observation. In addition to possible missing primary source emission data, the existence of currently undiscovered VOCs cannot be ruled out, which may act as a secondary source of formaldehyde through multiple $RO_2 + NO$ reaction steps (Färber et al., 2024).

[Figure]

**Fig. S11.** The observed and modeled HCHO concentration during the TROPSTECT-YRD campaign.

[revised manuscript text omitted]

under severe ozone pollution conditions.

**Minor Comments**

*1. Line 38: Define '*ChL*'*

**Reply:**

The relevant deception in the abstract has been deleted.

*2. Line 232 and 235: the different notations used in (2) and (3) need to be defined.*

**Reply:**

The relevant modifications have been added to Line 264-265.

**Revision:**

Line 264-265: Here, the OH yields from ozone photolysis and ozonolysis reactions are denoted as $\varphi_{OH}$ and $\varphi_{OH}^i$, respectively.

[revised manuscript text omitted]

---

## Author Comment (AC2)

Dear Editor Eleanor Browne and Referee,

Thanks for your suggestions which significantly help us to improve the manuscript. Hereby, we submit our responses and the manuscript has been revised accordingly. If there are any further questions or comments, please let us know.

Best regards

Guoxian Zhang on behalf of all co-authors

Key Lab. of Environmental Optics & Technology, Anhui Institute of Optics and Fine Mechanics, Chinese Academy of Sciences

230031 Hefei China

E-mail: gxzhang@aiofm.ac.cn

**Major Comments**

*1. Species X to match the observation. This is nothing more than a fitting exercise and does not really help us understanding what mechanism might be behind. Could be removed.*

**Reply:**

Thank you for your review and valuable comments on this manuscript. We agree with your perspective on Species X, considering its current role more as a fitting parameter, which may offer limited help in understanding the underlying mechanisms. However, in Section 3.3, we compared the oxidizing capacity of different urban agglomerations in China, using Species X as a comparative factor to demonstrate the extent of missing OH radical sources in various regions. Therefore, in this revision, we have decided to retain the discussion on Species X. Furthermore, to enhance the depth of the research and understanding of the mechanisms, we have incorporated the higher aldehyde mechanism (HAM) in subsequent studies and tested its impact on OH radicals, thereby complementing the discussion on Species X.

*2. Introduction of more monoterpenes which might sustain a lower-than-expected HO2 to RO2 ratio due to the chemistry of complex alkoxy radicals. This in the current version of the paper is not well explained though. How does the RACM-LIM1 treats the alkoxy radicals formed from alpha-pinene and limonene? Did the author modified the mechanisms including available SAR? How is the organic nitrate yield treated? A recent study by Färber et al. (2024) shows that it might be difficult to sustain a lower than 0.6 HO2-to-RO2 ratio due to termination reaction for complex RO2 such as formation of organic nitrates. The section in the paper showing the sensitivity test including monoterpenes should give more information.*

**Reply:**

Thank you for your reply. The oxidation processes of α-pinene and limonene related to RACM2 mechanism have been listed in Table S4, including the oxidation reactions with $OH/O_3/NO_3$, as well as the reactions of the derived alkoxy radicals

(APIP) with $NO/NO_3/HO_2$ and the self-reactions among peroxy radicals. In the RACM2 mechanism, the peroxy radicals generated from α-pinene oxidation are classified as APIP and return to $HO_2$ radicals through subsequent reactions with NO. This manuscript has not modified the mechanisms including the available structure-activity relationships (SAR), and the yield of organic nitrates still uses the results from the RACM2-LIM1 mechanism, which is specifically described in Table S4. The sensitivity testing section that includes monoterpenes has provided more information on Lines 535-555.

We followed the reviewers' suggestions and added a discussion on the $HO_2/RO_2$ ratio in Section 4.2. Regarding the $HO_2/RO_2$ ratio issue for experiments and simulations, we have summarized the $HO_2/RO_2$ radical concentration ratios derived from radical concentrations measured by laser-induced fluorescence instruments and calculated using the MCM or RACM mechanisms in Fig. 10. If $HO_2$ is formed from an $RO_2$ radical, it would result in an $HO_2/RO_2$ radical concentration ratio of approximately 1. In field studies, the observed $HO_2/RO_2$ ratios were between 0.2 - 1.7 under low-NO conditions (NO < 1 ppb) and only 0.1 - 0.8 under high-NO conditions (3 < NO < 6 ppb). From the perspective of model-observation matching, except for three measurements in ClearfLo, ICOZA and AIRPRO-summer campaigns, the $HO_2/RO_2$ ratios in other regions could be reasonably reflected by the MCM or RACM2 mechanisms. However, the ratio is generally underestimated under high NO conditions, reaching up to 5 times in ClearfLo. According to the latest chamber experiments, the $HO_2/RO_2$ radical concentration ratios for VOCs forming $HO_2$ are 0.6 for both one-step and two-step reactions. Therefore, the extremely low $HO_2/RO_2$ ratios observed in field campaigns can only be explained if almost all $RO_2$ radicals undergo multiple-step reactions before forming $HO_2$. During the TROPSTECT campaign, the observed $HO_2/RO_2$ remains at 1.1 and 0.8 under low-NO and high-NO conditions, respectively. After considering the complex sources of complex alkoxy radicals in the 'MTS+X' scenario, the simulated values of $HO_2/RO_2$ in both low-NO and high-NO regions match the observed values well.

[Figure]

**Fig. 10.** Summary of the HO$_2$/RO$_2$ ratios derived from radical concentrations measured by laser-induced fluorescence instruments and calculated using the MCM or RACM models under **(a)** low-NO and **(b)** high-NO conditions. Charmber Exp. 1 and Charmber Exp. 2 denotes the parameters by single-step HO$_2$ formation and multi-step HO$_2$ formation determined in the chamber by (Färber et al., 2024).

**Revision:**

Line 535-555: An additional reaction was added to the base model in a previous research, converting OH into C96O2 (the oxidation product of α-pinene) with a reaction rate equal to the missing reactivity, to explore the source of the missing RO$_2$ radicals(Whalley et al., 2021). Discrepancy of OH reactivity (~3 – 5 s$^{-1}$) between measurement and model suggested that an additional driving force was necessary to complete the OH to RO$_2$ step. In the TROPSPECT campaign, approximately 0.4 ppb of monoterpene was introduced into the base scenario as the chemical reactions of complex alkoxy radicals, which is similar to an atmospheric level in the EXPLORE-2018 campaign, the YRD region (Wang et al., 2022a). The RACM2 mechanism identified α-pinene (API) and limonene (LIM) as representative monoterpenes species. Sensitivity tests were conducted by incorporating API and LIM into the 'MTS on' and 'MTS+X on' scenarios, respectively (Ma et al., 2022). The mean of these values was considered the average effect of monoterpenes chemistry, and depicted as the green line in Fig. 6. In the 'MTS on' scenario, the chemistry of peroxy radicals in Semi II was reasonably described by introducing the source of complex alkoxy radicals, and the obs-to-mod ratio of peroxy radicals decreased from 2.2 to 1.3.

Furthermore, the introduction of additional complex alkoxy radicals had minimal impact on HOx chemistry, with changes in daytime OH and HO$_2$ concentrations of less than $5\times10^5$ cm$^{-3}$ and $2.5\times10^7$ cm$^{-3}$, respectively. This demonstrates the robustness of HOx radical in response to potential monoterpene.

Line 635-654: The HO$_2$/RO$_2$ parameter was utilized to explore the transformation relationship between HO$_2$ and RO$_2$ radicals. If HO$_2$ is formed from an RO$_2$ radical, it would result in an HO$_2$/RO$_2$ radical concentration ratio of approximately 1. The HO$_2$/RO$_2$ ratios derived from radical concentrations measured by laser-induced fluorescence instruments and calculated using the MCM or RACM models were summarized in Fig. 10. In field studies, the observed HO$_2$/RO$_2$ ratios were between 0.2 - 1.7 under low-NO conditions (NO < 1 ppb) and only 0.1 - 0.8 under high-NO conditions (3 < NO < 6 ppb). From the perspective of model-observation matching, except for three measurements in ClearfLo, ICOZA and AIRPRO-summer campaigns, the HO$_2$/RO$_2$ ratios in other regions could be reasonably reflected by the MCM or RACM2 mechanisms(Woodward-Massey et al., 2023; Whalley et al., 2021; Whalley et al., 2018; Färber et al., 2024). However, the ratio is generally underestimated under high NO conditions, reaching up to 5 times in ClearfLo. According to the latest chamber experiments, the HO$_2$/RO$_2$ radical concentration ratios for VOCs forming HO$_2$ are 0.6 for both one-step and two-step reactions. Therefore, the extremely low HO$_2$/RO$_2$ ratios observed in field campaigns can only be explained if almost all RO$_2$ radicals undergo multiple-step reactions before forming HO$_2$. During the TROPSTECT campaign, the observed HO$_2$/RO$_2$ remains at 1.1 and 0.8 under low-NO and high-NO conditions, respectively. After considering the complex sources of complex alkoxy radicals in the 'MTS+X' scenario, the simulated values of HO$_2$/RO$_2$ in both low-NO and high-NO regions match the observed values well.

3. *The last "manipulation" of the mechanisms is not really clear to me. In the text it is mentioned: "Manipulating the self-reaction rate of peroxy radicals by approximately five-fold, and the extended lifetime counterbalance their supplementary consumption by non-traditional regeneration mechanisms"*

*(Page 18 lines 465-467). I have no idea of what this means practically in the mechanism. This needs to be explained in a clearer way.*

**Reply:**

Thank you for your reply. The last 'manipulation' of the mechanisms is based on the 'MTS+X' scenario, aiming to test the impact of reducing the rate coefficients between peroxy radicals on the concentration of $RO_2$ radicals. We acknowledge that this part of the content has little connection with other sensitivity tests, therefore we have deleted this discussion and supplemented the relevant discussion on the impact of the HAM mechanism on $RO_2$ radicals.

**Table.1.** The sensitive test scenarios utilized to improve the model-measurement consistency between OH, $HO_2$ and $RO_2$ radicals.

| Scenario | Configuration | Purpose |
|---|---|---|
| Base | RACM2 updated with isoprene reaction scheme (LIM) | The base case with the species involved in Table S3 are constrained as boundary conditions. |
| X on | As the base scenario, but add the X mechanism, and the X level is between 0.25 - 0.5 ppb. | To untangle the missing OH source where base scenario failed. |
| MTS on | As the base scenario, but add a monoterpene source, and the monoterpene level is ~0.4 ppb. | Utilizing monoterpene-derived $RO_2$ to represent the alkoxy radicals with rather complex chemical structures. |
| MTS+X on | As the base scenario, but both the X mechanism and monoterpene source are considered. | To consider both the missing OH and $RO_2$ sources. |
| HAM on | As the base scenario, but add the reactive aldehyde chemistry. | To provide a test of whether the proposed mechanism can explain the missing OH source. |
| HAM on (4 × ALD) | As the base scenario, but add the reactive aldehyde chemistry, and the concentration of ALD was amplified by a factor of 4. | To quantify the impact of missing aldehyde primary emissions on ROx chemistry. |
| Ozone simulation | As the base scenario, but remove the constraints of the observed ozone and NO concentrations. | To test the suitable lifetime for the base model. |
| HCHO simulation | As the base scenario, but remove the constraint of the observed HCHO concentration. | To test the simulation effect of the existing mechanism on formaldehyde concentration. |

**Revision:**

Line 556-569: Higher aldehyde chemistry is a concrete manifestation of verifying the aforementioned hypothesis for $RO_2$ sources(Yang et al., 2024b). The autoxidation process of $R(CO)O_2$, encompasses a hydrogen migration process that transforms it

into the ·OOR(CO)OOH radical(Wang et al., 2019). This radical subsequently reacts with NO to yield the ·OR(CO)OOH radical. The ·OR(CO)OOH radical predominantly undergoes two successive rapid hydrogen migration reactionss, ultimately resulting in the formation of $HO_2$ radicals and hydroperoxy carbonyl (HPC). Consequently, the HAM mechanism extends the lifetime of the $RO_2$ radical, providing a valuable complement to the unaccounted sources of $RO_2$ radicals. As depicted in Fig. 7, the incorporation of the HAM mechanism results in an approximate 7.4% and 12.5% increase in the concentrations of $HO_2$ and $RO_2$ radicals, respectively. It is important to note that the total concentrations of primary emitted aldehydes and the HPC group may be underestimated, which could lead to the aforementioned analysis being conservative in nature. Further exploration of the unaccounted sources of $RO_2$ radicals will be presented in Section 4.3.

4. *As mentioned by Referee #1 many more details on how the model simulations are performed are needed. In the manuscript it is mentioned that species listed in table S1 are used to set the boundary conditions for the base scenario. Which NMHC are included? From the kOH budget it appears a large variety of different VOC was measured. It would be good to list them. Is the precision, accuracy and limit of detection the same for all the different VOCs and OVOCs measured? Focusing on the kOH budget plot I would recommend separating the contribution of HCHO (which I assume now is included in the OVOC label) as I would guess it might be the largest fraction of the OVOCs.*

**Reply:**

Thank you for your reply. We have listed the VOCs involved in the model simulation in Table S3.

**Table.S3.** The comprehensive list of model constraints.

| Categories | Species |
| --- | --- |
| Meteorology | Temperature, Relative humidity, Pressure, Jvalues |
| Trace gases | $O_3$, NO, $NO_2$, $SO_2$, CO, PAN, HONO |
| Alkanes | methane, ethane, propane, n-butane, isobutane, cyclopentane, n-pentane, isopentane, cyclohexane, methyl cyclopentane, 2,3-dimethyl butane, |

| | |
|---|---|
| | 2,2-dimethyl butane, n-hexane, 2-methyl pentane, 3-methyl pentane, methyl cyclohexane, n-heptane, 2-methyl hexane, 2,3-dimethyl pentane, 2,4-dimethyl pentane, 3-methyl hexane, n-octane, 2,3,4-trimethyl pentane, 2-methyl heptane, 3-methyl heptane, 2,2,4-trimethyl pentane, n-nonane, n-decane, n-undecane, n-dodecane |
| Alkenes | ethene, propene, 1,3-butadiene, 1-butene, cis-2-butene, trans-2-butene, 1-pentene, cis-2-pentene, trans-2-pentene, 1-hexene, styrene |
| BVOCs | isoprene |
| Alkynes | acetylene |
| Aromatics | benzene, toluene,ethyl benzene, o-xylene, m-xylene, n-propyl benzene, isopropyl benzene, p-ethyl toluene, o-ethyl toluene, m-ethyl toluene, 1,2,4-trimethyl benzene, 1,3,5-trimethyl benzene, 1,2,3-trimethyl benzene, p-diethyl benzene, m-diethyl benzene |
| OVOCs | HCHO, acetaldehyde, MACR, MVK |

The precision, accuracy, and detection limits for different VOCs and OVOCs measured using online GC-MS/FID are not the same, with some information on VOCs already listed in Table S2.

**Table S2.** Information table for parts of the VOC monitoring species by online GC-MS/FID. Revised by (Zhu et al., 2021).

| Name | Molecular formula | m/z | MIR | Uncertainty | LOD |
|---|---|---|---|---|---|
| MTBE | $C_5H_{12}O$ | 88.15 | 0.73 | 3.3% | 0.012 |
| Ethane | $C_2H_6$ | 30.07 | 0.28 | 4.6% | 0.013 |
| Propane | $C_3H_8$ | 44.10 | 0.49 | 0.9% | 0.010 |
| n-Butane | $C_4H_{10}$ | 58.12 | 1.15 | 0.3% | 0.012 |
| Isobutane | $C_4H_{10}$ | 58.12 | 1.23 | 0.6% | 0.008 |
| Isopentane | $C_5H_{12}$ | 72.15 | 1.45 | 0.7% | 0.008 |
| n-Pentane | $C_5H_{12}$ | 72.15 | 1.31 | 1.5% | 0.008 |
| Cyclohexane | $C_6H_{12}$ | 84.16 | 1.25 | 1.5% | 0.013 |
| n-Hexane | $C_6H_{14}$ | 86.18 | 1.24 | 2.0% | 0.006 |
| 2-Methylpentane | $C_6H_{14}$ | 86.18 | 1.5 | 3.8% | 0.009 |
| 3-Methylpentane | $C_6H_{14}$ | 86.18 | 1.8 | 1.9% | 0.009 |
| Ethylene | $C_2H_4$ | 28.05 | 9 | 1.5% | 0.013 |
| Propene | $C_3H_6$ | 42.08 | 11.66 | 1.0% | 0.010 |
| Acetylene | $C_2H_2$ | 26.04 | 0.95 | 1.3% | 0.018 |
| Chloromethane | $CH_3Cl$ | 50.49 | 0.038 | 9.1% | 0.011 |
| Dichloromethane | $CH_2Cl_2$ | 84.93 | 0.041 | 3.2% | 0.001 |
| 1,2-Dichloroethane | $C_2H_4Cl_2$ | 98.96 | 0.21 | 3.4% | 0.001 |
| 1,2-Dichloropropane | $C_3H_6Cl_2$ | 112.99 | 0.29 | 1.1% | 0.012 |
| Chloroform | $CHCl_3$ | 119.38 | 0.022 | 1.2% | 0.007 |
| Freon-11 | $CCl_3F$ | 137.40 | / | 4.6% | 0.010 |
| 1,3-Dichlorobenzene | $C_6H_4Cl_2$ | 147.00 | / | 9.6% | 0.022 |
| Tetrachloromethane | $CCl_4$ | 153.82 | 0 | 1.5% | 0.003 |
| Freon-113 | $C_2Cl_3F_3$ | 187.38 | / | 2.7% | 0.004 |

Regarding the distribution of $k_{OH}$, the contribution of HCHO has been separately

identified according to the reviewers' opinions, and a more detailed discussion has been conducted on the contribution of OVOCs to OH reactivity (Fig. 4).

[Figure]

**Fig. 4.** Timeseries of the observed and modelled parameters for OH, HO₂ and $k_{OH}$ during the observation period. **(a)** OH, **(b)** HO₂, **(c)** $k_{OH}$.

We added the description in Line 138-139&345-353&372-386.

**Revision:**

Line 139-140: Information table for parts of the VOC monitoring species by online GC-MS/FID was listed in Table S2.

Line 345-353: $k_{OVOCs}$ are categorized into three groups: $k_{OVOCs(Obs)}$, $k_{OVOCs(Model)}$, and $k_{HCHO}$. Given the significance of formaldehyde photolysis, the contribution of HCHO to $k_{OVOCs}$ is distinguished. $k_{OVOCs(Obs)}$ encompasses species observed in addition to formaldehyde, such as acetaldehyde (ACD) and the oxidation products of isoprene (MACR and MVK). Intermediates generated by the model, including glyoxal (GLY), methylglyoxal (MGLY), higher aldehydes (ALD), ketones (KET), methyl ethyl ketone (MEK), and methanol (MOH), are classified as $k_{OVOCs(Model)}$. Upon considering $k_{OVOCs(Model)}$, the reactivity calculated prior to September 10th aligns quite well with the observed OH reactivity.

Line 372-386: The calculated reactivity seems to compare well with the observed OH

reactivity at the start of the measurement period, but then there is evidence of missing OH reactivity after September 10th (Fig. 4(d)). Due to the limitations of available instruments, this observation only measured a limited number of OVOCs species, making it difficult to accurately quantify the contribution of larger aldehydes and ketones, carboxylic acids, nitrophenols, and other multifunctional species to $k_{OH}$ (Wang et al., 2024). Since the MCM mechanism considers more secondary formation reactions than the RACM2 mechanism, it can qualitatively assess the photochemical role of unmeasured OVOCs species in the atmosphere (Wang et al., 2022b). The additional modeled OVOCs by the MCM v3.3.1 mechanism contributed ~2.4 s$^{-1}$ to the missing OH reactivity (Fig. S7). During Heavy period, the reactivity of more model oxidation products increased the daytime $k_{OH}$ by about 5.1 s$^{-1}$. Therefore, the observed $k_{OH}$ can serve as an upper limit for sensitivity tests, thereby the full suite of radical measurement can be performed to explore the missing oxidation properties and ozone formation (Section 4.1).

5. *It would be good to add the experimental budget for ROx as looking at table S1, all the species contributing substantially in the modelled budget (Fig 5) are measured. Or is Fig. 5 showing the experimental budget? And why did the author only analysis ROx and not OH, HO2 and RO2 separately?*

**Reply:**

Thank you for your reply. We have added the experimental budget for OH, HO$_2$, RO$_2$ and the total ROx according to the reviewers' suggestions. We have newly added Section 2.4 which details the relevant methods for calculation, and have supplemented the experimental budget results in Fig. S8.

[Figure]

**Fig. S8.** Experimental budget for OH, HO₂, RO₂ and total ROx radicals during different periods.

**Revision:**

Line 269: 2.4 Experimental budget analysis

Line 270-284: In this study, an experimental radical budget analysis was also conducted (Eqs. (5) - (12)). Unlike model studies, this method relies solely on field measurements (concentrations and photolysis rates) and published chemical kinetic data, without depending on concentrations calculated by models(Whalley et al., 2021; Tan et al., 2019). Given the short-lived characteristics of OH, HO₂, and RO₂ radicals, it is expected that the concentrations are in a steady state, with total production and loss rates being balanced(Lu et al., 2019a). By comparing the known sources and sinks for radicals, unknown processes for initiation, transformation and termination can be determined.

$$P(OH) = j_{HONO}[HONO] + \varphi_{OH} j(O^1D)[O_3] + \Sigma i \left\{ \varphi_{OH}^i k_{Alkenes+O_3}^i [Alkenes][O_3] \right\}$$

$$+ (k_{HO_2+NO}[NO] + k_{HO_2+O_3}[O_3])[HO_2] \tag{6}$$

$$D(OH) = [OH] \times k_{OH} \tag{7}$$

$$P(HO_2) = 2 \times j_{HCHO\_R}[HCHO] + \Sigma i \left\{ \varphi^i_{HO_2} k^i_{Alkenes+O_3}[Alkenes][O_3] \right\}$$

$$+ (k_{HCHO+OH}[HCHO] + k_{CO+OH}[CO])[OH]$$

$$+ \alpha k_{RO_2+NO}[NO][RO_2] \tag{8}$$

$$D(HO_2) = (k_{HO_2+NO}[NO] + k_{HO_2+O_3}[O_3] + k_{HO_2+RO_2}[RO_2]$$

$$+ 2 \times k_{HO_2+HO_2}[HO_2])[HO_2] \tag{9}$$

$$P(RO_2) = \Sigma i \left\{ \varphi^i_{RO_2} k^i_{Alkenes+O_3}[Alkenes][O_3] \right\}$$

$$+ k_{OH}[VOCs][OH] \tag{10}$$

$$D(RO_2) = \left\{ (\alpha + \beta)k_{RO_2+NO}[NO] + (2 \times k_{RO_2+RO_2}[RO_2] \right.$$

$$+ k_{HO_2+RO_2}[HO_2])\}[RO_2] \tag{11}$$

$$P(RO_x) = \Sigma i \left\{ (\varphi^i_{OH} + \varphi^i_{HO_2} + \varphi^i_{RO_2})k^i_{Alkenes+O_3}[Alkenes][O_3] \right\} + j_{HONO}[HONO]$$

$$+ \varphi_{OH} j(O^1D)[O_3] + 2 \times j_{HCHO\_R}[HCHO] \tag{12}$$

$$D(RO_x) = (k_{OH+NO_2}[NO_2] + k_{OH+NO}[NO])[OH] + \beta k_{RO_2+NO}[NO]$$

$$+ 2 \times (k_{RO_2+RO_2}[RO_2][RO_2] + k_{HO_2+RO_2}[HO_2][RO_2]$$

$$+ k_{HO_2+HO_2}[HO_2][HO_2]) \tag{13}$$

In which, j(HONO), j(O$^1$D) are the measured photolysis rates of HONO and O$_3$, respectively, and jHCHO_R is the measured photolysis rate for the channel of formaldehyde photolysis generating HO$_2$. $\varphi_{OH}$ represent the OH yield in the O$_3$ photolysis reaction. $\varphi^i_{OH}$, $\varphi^i_{HO_2}$ and $\varphi^i_{RO_2}$ are the yields for the ozonolysis reaction producing OH, HO$_2$, and RO$_2$, respectively. α is the proportion of RO$_2$ radicals reacting with NO that are converted to HO$_2$, and β is the proportion of alkyl nitrates formation, which are set to 1 and 0.05, respectively(Tan et al., 2019).

Line 407-433: By comparing the known sources and sinks for radicals, unknown processes for initiation, transformation and termination can be determined in the experimental budget analysis (Fig. S8). During the Semi I period, the production and destruction rates of HO$_2$, RO$_2$, and total ROx radicals were very consistent, but a significant lack of a source term for OH radicals was existed after 10:00. This missing source became more pronounced during the Heavy period, reaching 16 ppb/h at noon,

which is close to the results observed by AIRPRO, but three times that observed by Heshan in PRD region(Tan et al., 2019; Whalley et al., 2021). The ratio of OH production-to-destruction rate during the Semi II period was close to 1, indicating consistency between the observed results of OH, $HO_2$, $k_{OH}$, and other precursors(Whalley et al., 2018). However, the generation of $HO_2$ radicals in the morning was about twice as high as the removal rate, suggesting that there are contributions from unconsidered $HO_2$ radical removal channels (such as heterogeneous reactions)(Song et al., 2021). During the Heavy period, there was a rapid total removal rate of $RO_2$ radicals, reflecting the dominated $HO_2$ generation by the reaction of $RO_2$ radicals with NO. Although the $P(HO_2)$ and $D(HO_2)$ were quite in balance, the removal rate of $RO_2$ radicals far exceeded the known production rate (especially before 12:00). Previous work has shown that halogen chemistry (such as photolysis of nitryl chloride ($ClNO_2$)) could be an important source in the morning time, but this was not included in the calculation of ROx or $RO_2$ budget in this campaign. The steady-state analysis for $HO_2$ radical in the London campaign emphasized that only by significantly reducing the observed $RO_2$-to-$HO_2$ propagation rate to just 15% could balance both $P(HO_2)$ and $D(HO_2)$, indicating that the $RO_2$-related mechanism for propagation to other radical species may not be fully understood(Whalley et al., 2018). Therefore, based on the current knowledge seems unlikely to explain the required source-sink difference of nearly 25 ppb/h in the $RO_2$ budget. Sensitivity analysis is needed to further infer the causes of the difference for the experimental budget analysis.

6. *Co-authors of this study just recently published a new mechanisms that could explain the missing OH source at low NO (Yang et al., 2024). This could be a good sensitivity test rather than species X and I would recommend the authors to try it.*

**Reply:**

Thank you for your review and valuable comments on this manuscript. We agree with your perspective on species X, considering that it currently serves more as a

fitting parameter, which may offer limited assistance in understanding the underlying mechanisms. Therefore, in this revision, we have followed your advice and added the Higher Aldehyde Mechanism to test whether it can explain the discrepancy between measured and simulated radical concentrations. The results indicate that the contribution of the HAM mechanism to OH radicals in different episodes ranged between 4.4% - 6.0%, while the concentrations of $HO_2$ and $RO_2$ radicals increased by approximately 7.4% and 12.5%, respectively.

[Figure]

**Fig. 7.** The response of **(a)** OH, **(b)** $HO_2$ and **(c)** $RO_2$ radicals to the Higher Aldehyde Mechanism (HAM) in different episodes (Semi I, Heavy, and Semi II) in diurnal time (10:00-15:00).

Additionally, we combine the missing aldehyde primary emissions and the HAM mechanism under the entire photochemical spectrum to qualitatively assess the impact on $RO_2$ radical concentrations. Notably, $RO_2$ radical concentrations exhibit a pronounced sensitivity to autoxidation, with the incorporation of additional OVOCs potentially boosting simulated $RO_2$ radical concentrations by 20% to 40%. Consequently, although limiting formaldehyde can partially offset the $HO_2$ radical cycle and enhance the precision of HOx radical chemistry studies, additional measurements should be undertaken for other OVOCs, coupled with the deployment of full-chain radical detection systems, to accurately elucidate the oxidation processes under severe ozone pollution conditions.

[Figure]

**Fig. S12.** The relationship between the differences in the simulation of **(a)** OH, **(b)** $HO_2$, and **(c)** $RO_2$ radical concentrations by HAM mechanism and the base scenario across the entire photochemical spectrum. An empirical hypothesis is proposed to amplify the concentration of higher-order aldehydes by a factor of about 4, which is the proportion of formaldehyde concentration underestimated by the model. The boxplots represent the 10%, 25%, median, 75%, and 90% of the data, respectively.

**Revision:**

Line 508-518: Missing OH sources are closely related to the chemistry of OVOCs(Yang et al., 2024a; Qu et al., 2021). Reactive aldehyde chemistry, particularly the autoxidation of carbonyl organic peroxy radicals ($R(CO)O_2$) derived from higher aldehydes, is a significant OH regeneration mechanism that has been shown to contribute importantly to OH sources in regions with abundant natural and anthropogenic emissions during warm seasons(Yang et al., 2024b). In this study, the higher aldehyde mechanism (HAM) by Yang et al was parameterized into the base model to test new insights into the potential missing radical chemistry (Fig. 7). The results indicate that the contribution of the HAM mechanism to OH radicals in different episodes ranged between 4.4% - 6.0%, while the concentrations of $HO_2$ and $RO_2$ radicals increased by approximately 7.4% and 12.5%, respectively.

Line 556-569: Higher aldehyde chemistry is a concrete manifestation of verifying the aforementioned hypothesis for $RO_2$ sources(Yang et al., 2024b). The autoxidation

process of $R(CO)O_2$, encompasses a hydrogen migration process that transforms it into the $\cdot OOR(CO)OOH$ radical(Wang et al., 2019). This radical subsequently reacts with NO to yield the $\cdot OR(CO)OOH$ radical. The $\cdot OR(CO)OOH$ radical predominantly undergoes two successive rapid hydrogen migration reactionss, ultimately resulting in the formation of $HO_2$ radicals and hydroperoxy carbonyl (HPC). Consequently, the HAM mechanism extends the lifetime of the $RO_2$ radical, providing a valuable complement to the unaccounted sources of $RO_2$ radicals. As depicted in Fig. 7, the incorporation of the HAM mechanism results in an approximate 7.4% and 12.5% increase in the concentrations of $HO_2$ and $RO_2$ radicals, respectively. It is important to note that the total concentrations of primary emitted aldehydes and the HPC group may be underestimated, which could lead to the aforementioned analysis being conservative in nature. Further exploration of the unaccounted sources of $RO_2$ radicals will be presented in Section 4.3.

Line 682-712: The reasons for the discrepancy between simulated and observed values for ozone production deserve further investigation. As depicted in Fig.11(c), the simulated $HO_2/RO_2$ ratios display a robust positive correlation with photochemical activity, fluctuating between 2 and 4. A notable feature during severe ozone pollution is the intense distribution of formaldehyde, with an average concentration of $21.81 \pm 4.57$ ppb (11:00 – 13:00). While formaldehyde acts as a precursor for $HO_2$ radicals, it does not directly generate $RO_2$ radicals. The contributions of OVOCs to the ROx radical do not exhibit the same intensity as formaldehyde, and the current mechanism encounters difficulties in replicating formaldehyde concentrations (Fig. S11). The simulation of formaldehyde concentrations using the MCM v3.3.1 mechanism has shown improvement, indicating that the secondary formation of unmeasured species, such as OVOCs, will feedback on $RO_2$ radical levels. When formaldehyde levels are unconstrained, the simulated $HO_2/RO_2$ ratios align with observations, suggesting that under the prevailing chemical mechanism, the photochemical efficiency of formaldehyde and other OVOCs is similar. Therefore, an empirical hypothesis is proposed to amplify the concentration of higher-order aldehydes by a factor of about 4, which is the proportion of formaldehyde concentration underestimated by the model.

The qualitative assessment of the impact of missing aldehyde primary emissions on $RO_2$ radical concentrations was combined with the HAM mechanism across the entire photochemical spectrum (Fig.S12). Enhanced impact of aldehyde autoxidation in the presence of weak photochemical conditions could alter the simulated levels of OH and $HO_2$ radicals by approximately 13.9% and 18.1%, respectively. However, higher ALD concentrations will be achieved under intensive photochemical conditions, leading to the gradual dominance of the sink channels for OH + OVOCs, with the effect of autoxidation mechanisms gradually decreasing. $RO_2$ radical concentrations is notably more sensitive to the HAM mechanism, where incorporates additional OVOCs, can enhance the simulation of $RO_2$ radical concentrations by 20 - 40%. Consequently, although limiting formaldehyde can partially offset the $HO_2$ radical cycle and enhance the precision of HOx radical chemistry studies, additional measurements should be undertaken for other OVOCs, coupled with the deployment of full-chain radical detection systems, to accurately elucidate the oxidation processes under severe ozone pollution conditions.

**Minor Comments**

1. *"The full-chain radical detection untangled a gap-bridge between the photochemistry and the intensive oxidation level in the chemical-complex atmosphere, enabling a deeper understanding of the tropospheric radical chemistry at play."* (Page 2, Lines 42-45)

**Reply:**

Thank you for your suggestion, the abstract section has been re-optimized.

**Revision:**

Line 30-48: At a heavy ozone pollution episode, the oxidation capacity reached an intensive level compared with other sites, and the simulated OH, $HO_2$, and $RO_2$ radicals provided by the RACM2-LIM1 mechanism failed to adequately match the observed data both in radical concentration and experimental budget analysis. Sensitivity tests utilizing a comprehensive set of radical measurements revealed that the higher aldehyde mechanism (HAM) effectively complements the non-traditional regeneration of OH radicals, yielding enhancements of 4.4% - 6.0% compared to the base scenario, while the concentrations of $HO_2$ and $RO_2$ radicals have shown increments of about 7.4% and 12.5%, respectively. Notably, $RO_2$ radical concentrations exhibit a pronounced sensitivity to autoxidation, with the incorporation of additional OVOCs potentially boosting simulated $RO_2$ radical concentrations by 20% to 40%. The incorporation of larger alkoxy radicals stemming from monoterpenes has refined the consistency between measurements and modeling in the context of ozone production under elevated NO levels, diminishing the disparity from 4.17 to 2.33. This outcome corroborates the hypothesis of sensitivity analysis as it pertains to ozone formation. Moving forward, by implementing a comprehensive radical detection approach, further investigations should concentrate on a broader range of OVOCs to rectify the imbalance associated with $RO_2$ radicals, thereby providing a more precise understanding of oxidation processes during severe ozone pollution episodes.

2. *"Moreover, the closure experiment, incorporating field campaigns and box*

*model, has proven to be an effective method for verifying the integrity of radical chemistry at local to global scales.* ” *(Page 3, Lines 70-72). I do not know what the closure experiment is?*

**Reply:**

Thank you for your reply, we have revised the relevant description in Line 73-75.

**Revision:**

Line 73-75: Moreover, the union of comprehensive field campaigns and box model, has proven to be an effective method for verifying the integrity of radical chemistry at local to global scales (Lu et al., 2019b; Tan et al., 2018).

**References**

[revised manuscript text omitted]

Zhu, B., Huang, X.-F., Xia, S.-Y., Lin, L.-L., Cheng, Y., and He, L.-Y.: Biomass-burning emissions could significantly enhance the atmospheric oxidizing capacity in continental air pollution, Environ. Pollut., 285, 10.1016/j.envpol.2021.117523, 2021.

---

## Author Comment (AC3)

Dear Editor Eleanor Browne and Referee,

Thanks for your suggestions which significantly help us to improve the manuscript. Hereby, we submit our responses and the manuscript has been revised accordingly. If there are any further questions or comments, please let us know.

Best regards

Guoxian Zhang on behalf of all co-authors

Key Lab. of Environmental Optics & Technology, Anhui Institute of Optics and Fine Mechanics, Chinese Academy of Sciences

230031 Hefei China

E-mail: gxzhang@aiofm.ac.cn

**Major Comments**

*1. The authors did not conduct any testing for potential interferences associated with their OH measurements. While it is clear that some LIF-FAGE instruments are more sensitive to interferences than others, testing for interferences is still important, especially in complex chemical environments given that the source(s) of the interference have yet to be identified. The authors should acknowledge the possibility that unknown interferences may have contributed to their OH measurements and may explain some of the discrepancy with their model. It is unfortunate that the authors did not conduct interference testing during the "heavy" pollution episode. This would have provided confidence that the elevated OH concentrations during this period were free from interferences.*

**Reply:**

Thanks for your suggestion. During the TROPSTECT-YRD campaign, we did not use an inlet-pre-injector to determine the chemical background of OH radical. We acknowledge your point that the comparison exercise with a second LIF instrument at a different location does not ensure that the instrument (and the OH measurement presented here) is free from interferences. We will discuss whether internal interference exists in AIOFM-LIF from the following aspects:

First of all, literature research shows that measurement interference is more related to the length of the inlet in the low-pressure cell (Griffith et al., 2016). In terms of system design, the AIOFM-LIF system uses a short-length inlet design to minimize this and other unknown disturbances (the distance from radical sampling to flourescence excitation is ~150 mm).

Additionally, potential interference may exist when the atmosphere contains abundant alkenes, ozone, and BVOCs, indicating that environmental conditions play leading roles in OH interferences (Mao et al., 2010; Fuchs et al., 2016; Novelli et al., 2014). In the previous comparison exercise with a LIF instrument deployed an inlet pre-injector (PKU-LIF), the ozonolysis interference on the measurement consistency

of both systems was excluded under high-VOCs conditions (Zhang et al., 2022).

We have compared the chemical conditions during the intercomparison experiment and the current environmental conditions. Overall, the key parameters related to ozonolysis reactions ($O_3$、alkenes、isoprene and NOx) in TROPSTECT-YRD were similar to those during the comparison experiment, which is not conducive to generating potential OH interference.

**Table.** Comparison of key parameters related to ozonolysis reactions ($O_3$、 alkenes、 isoprene and NOx) between TROPSTECT-YRD and the intercomparison experiment. All the values are the diurnal average (10:00-15:00).

| Species | Intercomparison | TROPSTECT-YRD |
|---|---|---|
| $O_3$ (ppb) | 71.02 | 76.25 |
| Alkenes (ppb) | 1.29 | 0.67 |
| Isoprene (ppb) | 0.67 | 0.86 |
| NOx (ppb) | 5.65 | 6.55 |

To provide direct evidence on the OH chemical background signal, we conducted another atmospheric oxidation observation in the same location (Science Island background station  in Hefei) and season (September, Autumn in 2022) in 2022, using chemical modulation methods to measure the chemical background of OH radicals in AIOFM-LIF instrument. The environmental conditions during ozone pollution (2022.9.29-2022.10.3) are shown in the Fig. S3, with daytime peaks of ozone concentration above 75 ppb, accompanied by alkene species approaching ~10 ppb. The diurnal concentration of isoprene was also a high level (>1 ppb). The chemical conditions are more favourable to induce OH interference than the TROPSTECT-YRD site. However, the OH concentrations achieved by chemical modulation ($OH_{chem}$) and wavelength modulation ($OH_{wav}$) were in good agreement. No obvious chemical background was observed by deploying an inlet pre-injector. Therefore, it is not expected that OH measurement in the present study was affected by internal interference.

[Figure]

**Fig. S3.** Results of an additional atmospheric oxidation observation experiment in the same location and season in 2022. **(a)** Ozone concentration **(b)** Concentrations of alkene and isoprene, respectively. **(c)** The OH concentrations achieved by chemical modulation ($OH_{chem}$) and wavelength modulation ($OH_{wav}$).

We added the detailed description in Line 187-197.

**Revision:**

Line 187-197: An additional atmospheric oxidation observation was conducted in the same location and season in 2022 with a chemical modulation method to determine the chemical background of OH radicals (Fig. S3). During the ozone pollution (2022.9.29-2022.10.3), the daytime peaks of ozone concentration above 75 ppb, accompanied by alkene species approaching ~10 ppb. The diurnal concentration of isoprene was also a high level (>1 ppb). The chemical conditions are more favourable to induce OH interference than in the TROPSTECT campaign, while the OH concentrations achieved by chemical modulation ($OH_{chem}$) and wavelength modulation ($OH_{wav}$) were in good agreement. No obvious chemical background was observed by deploying an inlet pre-injector. Therefore, it is not expected that OH measurement in the present study was affected by internal interference.

*2. There is very little discussion of the OH reactivity measurements. Figure 4 shows the measured reactivity with that calculated from major OH sinks, but it*

*isn'*t *clear whether these are the measured OH sinks or whether they include the reactivity of unmeasured modeled oxidation products. During the* "*heavy*" *pollution episode, the calculated reactivity appears to be higher than that during the* "*semi*" *periods, while the measured reactivity appears to be similar. Given that the greatest discrepancy between the radical measurements and the model occurred during the* "*heavy*" *episode, the modeled OH reactivity (including the reactivity of unmeasured modeled oxidation products) should be discussed in much more detail.*

**Reply:**

Thanks for your suggestion. First, we provided a detailed description of the $k_{OH}$ measurement instruments and listed the VOCs involved in the model simulations in Table S3.

**Table.S3.** The comprehensive list of model constraints.

| Categories | Species |
|---|---|
| Meteorology | Temperature, Relative humidity, Pressure, Jvalues |
| Trace gases | $O_3$, NO, $NO_2$, $SO_2$, CO, PAN, HONO |
| Alkanes | methane, ethane, propane, n-butane, isobutane, cyclopentane, n-pentane, isopentane, cyclohexane, methyl cyclopentane, 2,3-dimethyl butane, 2,2-dimethyl butane, n-hexane, 2-methyl pentane, 3-methyl pentane, methyl cyclohexane, n-heptane, 2-methyl hexane, 2,3-dimethyl pentane, 2,4-dimethyl pentane, 3-methyl hexane, n-octane, 2,3,4-trimethyl pentane, 2-methyl heptane, 3-methyl heptane, 2,2,4-trimethyl pentane, n-nonane, n-decane, n-undecane, n-dodecane |
| Alkenes | ethene, propene, 1,3-butadiene, 1-butene, cis-2-butene, trans-2-butene, 1-pentene, cis-2-pentene, trans-2-pentene, 1-hexene, styrene |
| BVOCs | isoprene |
| Alkynes | acetylene |
| Aromatics | benzene, toluene,ethyl benzene, o-xylene, m-xylene, n-propyl benzene, isopropyl benzene, p-ethyl toluene, o-ethyl toluene, m-ethyl toluene, 1,2,4-trimethyl benzene, 1,3,5-trimethyl benzene, 1,2,3-trimethyl benzene, p-diethyl benzene, m-diethyl benzene |
| OVOCs | HCHO, acetaldehyde, MACR, MVK |

Accordingly, we detailed the contribution of OVOCs to OH reactivity and analyzed the reasons for the differences between calculated and observed values (Fig.4). In Fig. 4, $k_{OVOCs}$ are categorized into three groups: $k_{OVOCs(Obs)}$, $k_{OVOCs(Model)}$,

and $k_{HCHO}$. Given the significance of formaldehyde photolysis, the contribution of HCHO to $k_{OVOCs}$ is distinguished. $k_{OVOCs(Obs)}$ encompasses species observed in addition to formaldehyde, such as acetaldehyde (ACD) and the oxidation products of isoprene (MACR and MVK). Intermediates generated by the model, including glyoxal (GLY), methylglyoxal (MGLY), higher aldehydes (ALD), ketones (KET), methyl ethyl ketone (MEK), and methanol (MOH), are classified as $k_{OVOCs(Model)}$. Upon considering $k_{OVOCs(Model)}$, the calculated reactivity seems to compare well with the observed OH reactivity at the start of the measurement period, but then there is evidence of missing OH reactivity after September 10th (Fig.4(d)).

[Figure]

**Fig. 4.** Timeseries of the observed and modelled parameters for OH, HO$_2$ and $k_{OH}$ during the observation period. **(a)** OH, **(b)** HO$_2$, **(c)** $k_{OH}$.

Due to the limitations of available instruments, this observation only measured a limited number of OVOCs species, making it difficult to accurately quantify the contribution of larger aldehydes and ketones, carboxylic acids, nitrophenols, and other multifunctional species to $k_{OH}$ (Wang et al., 2024). Since the MCM mechanism considers more secondary formation reactions than the RACM2 mechanism, it can qualitatively assess the photochemical role of unmeasured OVOCs species in the atmosphere (Wang et al., 2022). The additional modeled OVOCs by the MCM v3.3.1

mechanism contributed ~2.4 s⁻¹ to the missing OH reactivity (Fig.S6). During Heavy period, the reactivity of more model oxidation products increased the daytime $k_{OH}$ by about 5.1 s⁻¹. Therefore, the observed $k_{OH}$ can serve as an upper limit for sensitivity tests, thereby the full suite of radical measurement can be performed to explore the missing oxidation properties and ozone formation (Section 4.1).

[Figure]

**Fig. S7.** Timeseries of the observed and modelled $k_{OH}$ during the observation period.

We added the detailed description in Line 345-353&372-386.

**Revision:**

Line 345-353: $k_{OVOCs}$ are categorized into three groups: $k_{OVOCs(Obs)}$, $k_{OVOCs(Model)}$, and $k_{HCHO}$. Given the significance of formaldehyde photolysis, the contribution of HCHO to $k_{OVOCs}$ is distinguished. $k_{OVOCs(Obs)}$ encompasses species observed in addition to formaldehyde, such as acetaldehyde (ACD) and the oxidation products of isoprene (MACR and MVK). Intermediates generated by the model, including glyoxal (GLY), methylglyoxal (MGLY), higher aldehydes (ALD), ketones (KET), methyl ethyl ketone (MEK), and methanol (MOH), are classified as $k_{OVOCs(Model)}$. Upon considering $k_{OVOCs(Model)}$, the reactivity calculated prior to September 10th aligns quite well with the observed OH reactivity.

Line 372-386: The calculated reactivity seems to compare well with the observed OH reactivity at the start of the measurement period, but then there is evidence of missing OH reactivity after September 10th (Fig. 4(d)). Due to the limitations of available instruments, this observation only measured a limited number of OVOCs species, making it difficult to accurately quantify the contribution of larger aldehydes and ketones, carboxylic acids, nitrophenols, and other multifunctional species to $k_{OH}$

(Wang et al., 2024). Since the MCM mechanism considers more secondary formation reactions than the RACM2 mechanism, it can qualitatively assess the photochemical role of unmeasured OVOCs species in the atmosphere (Wang et al., 2022). The additional modeled OVOCs by the MCM v3.3.1 mechanism contributed ~2.4 s$^{-1}$ to the missing OH reactivity (Fig. S7). During Heavy period, the reactivity of more model oxidation products increased the daytime $k_{OH}$ by about 5.1 s$^{-1}$. Therefore, the observed $k_{OH}$ can serve as an upper limit for sensitivity tests, thereby the full suite of radical measurement can be performed to explore the missing oxidation properties and ozone formation (Section 4.1).

3. *As noted in the manuscript, there have been several studies where the "X mechanism" has been incorporated in order to explain the underprediction of the measured OH concentration by the model (Table 1). However, similar to these previous studies, the authors do not provide any new insight on what "X" might be. Some of these authors have recently published a theoretical study suggesting that reactive aldehyde chemistry may explain the missing source of OH (Yang et al., Nature Communications, 15, 1648 (2024). https://doi.org/10.1038/s41467-024-45885-w). Incorporation of this proposed mechanism into their model would provide some new insights into the potential missing radical chemistry and provide a test of whether the proposed mechanism can explain the measured radical concentrations during the heavy pollution episode.*

**Reply:**

Thank you for your review and valuable comments on this manuscript. We agree with your perspective on species X, considering that it currently serves more as a fitting parameter, which may offer limited assistance in understanding the underlying mechanisms. Therefore, in this revision, we have followed your advice and added the Higher Aldehyde Mechanism to test whether it can explain the discrepancy between measured and simulated radical concentrations. The results indicate that the

contribution of the HAM mechanism to OH radicals in different episodes ranged between 4.4% - 6.0%, while the concentrations of $HO_2$ and $RO_2$ radicals increased by approximately 7.4% and 12.5%, respectively.

[Figure]

**Fig. 7.** The response of **(a)** OH, **(b)** $HO_2$ and **(c)** $RO_2$ radicals to the Higher Aldehyde Mechanism (HAM) in different episodes (Semi I, Heavy, and Semi II) in diurnal time (10:00-15:00).

Additionally, we combine the missing aldehyde primary emissions and the HAM mechanism under the entire photochemical spectrum to qualitatively assess the impact on $RO_2$ radical concentrations. Notably, $RO_2$ radical concentrations exhibit a pronounced sensitivity to autoxidation, with the incorporation of additional OVOCs potentially boosting simulated $RO_2$ radical concentrations by 20% to 40%. Consequently, although limiting formaldehyde can partially offset the $HO_2$ radical cycle and enhance the precision of HOx radical chemistry studies, additional measurements should be undertaken for other OVOCs, coupled with the deployment of full-chain radical detection systems, to accurately elucidate the oxidation processes under severe ozone pollution conditions.

[Figure]

**Fig. S12.** The relationship between the differences in the simulation of **(a)** OH, **(b)** HO$_2$, and **(c)** RO$_2$ radical concentrations by HAM mechanism and the base scenario across the entire photochemical spectrum. An empirical hypothesis is proposed to amplify the concentration of higher-order aldehydes by a factor of about 4, which is the proportion of formaldehyde concentration underestimated by the model. The boxplots represent the 10%, 25%, median, 75%, and 90% of the data, respectively.

**Revision:**

Line 508-518: Missing OH sources are closely related to the chemistry of OVOCs(Yang et al., 2024a; Qu et al., 2021). Reactive aldehyde chemistry, particularly the autoxidation of carbonyl organic peroxy radicals (R(CO)O$_2$) derived from higher aldehydes, is a significant OH regeneration mechanism that has been shown to contribute importantly to OH sources in regions with abundant natural and anthropogenic emissions during warm seasons(Yang et al., 2024b). In this study, the higher aldehyde mechanism (HAM) by Yang et al was parameterized into the base model to test new insights into the potential missing radical chemistry (Fig. 7). The results indicate that the contribution of the HAM mechanism to OH radicals in different episodes ranged between 4.4% - 6.0%, while the concentrations of HO$_2$ and RO$_2$ radicals increased by approximately 7.4% and 12.5%, respectively.

Line 556-569: Higher aldehyde chemistry is a concrete manifestation of verifying the aforementioned hypothesis for RO$_2$ sources(Yang et al., 2024b). The autoxidation

process of R(CO)O$_2$, encompasses a hydrogen migration process that transforms it into the ·OOR(CO)OOH radical(Wang et al., 2019). This radical subsequently reacts with NO to yield the ·OR(CO)OOH radical. The ·OR(CO)OOH radical predominantly undergoes two successive rapid hydrogen migration reactionss, ultimately resulting in the formation of HO$_2$ radicals and hydroperoxy carbonyl (HPC). Consequently, the HAM mechanism extends the lifetime of the RO$_2$ radical, providing a valuable complement to the unaccounted sources of RO$_2$ radicals. As depicted in Fig. 7, the incorporation of the HAM mechanism results in an approximate 7.4% and 12.5% increase in the concentrations of HO$_2$ and RO$_2$ radicals, respectively. It is important to note that the total concentrations of primary emitted aldehydes and the HPC group may be underestimated, which could lead to the aforementioned analysis being conservative in nature. Further exploration of the unaccounted sources of RO$_2$ radicals will be presented in Section 4.3.

Line 682-712: The reasons for the discrepancy between simulated and observed values for ozone production deserve further investigation. As depicted in Fig.11(c), the simulated HO$_2$/RO$_2$ ratios display a robust positive correlation with photochemical activity, fluctuating between 2 and 4. A notable feature during severe ozone pollution is the intense distribution of formaldehyde, with an average concentration of 21.81 ± 4.57 ppb (11:00 – 13:00). While formaldehyde acts as a precursor for HO$_2$ radicals, it does not directly generate RO$_2$ radicals. The contributions of OVOCs to the ROx radical do not exhibit the same intensity as formaldehyde, and the current mechanism encounters difficulties in replicating formaldehyde concentrations (Fig. S11). The simulation of formaldehyde concentrations using the MCM v3.3.1 mechanism has shown improvement, indicating that the secondary formation of unmeasured species, such as OVOCs, will feedback on RO$_2$ radical levels. When formaldehyde levels are unconstrained, the simulated HO$_2$/RO$_2$ ratios align with observations, suggesting that under the prevailing chemical mechanism, the photochemical efficiency of formaldehyde and other OVOCs is similar. Therefore, an empirical hypothesis is proposed to amplify the concentration of higher-order aldehydes by a factor of about 4, which is the proportion of formaldehyde concentration underestimated by the model.

The qualitative assessment of the impact of missing aldehyde primary emissions on $RO_2$ radical concentrations was combined with the HAM mechanism across the entire photochemical spectrum (Fig.S12). Enhanced impact of aldehyde autoxidation in the presence of weak photochemical conditions could alter the simulated levels of OH and $HO_2$ radicals by approximately 13.9% and 18.1%, respectively. However, higher ALD concentrations will be achieved under intensive photochemical conditions, leading to the gradual dominance of the sink channels for OH + OVOCs, with the effect of autoxidation mechanisms gradually decreasing. $RO_2$ radical concentrations is notably more sensitive to the HAM mechanism, where incorporates additional OVOCs, can enhance the simulation of $RO_2$ radical concentrations by 20 - 40%. Consequently, although limiting formaldehyde can partially offset the $HO_2$ radical cycle and enhance the precision of HOx radical chemistry studies, additional measurements should be undertaken for other OVOCs, coupled with the deployment of full-chain radical detection systems, to accurately elucidate the oxidation processes under severe ozone pollution conditions.

*4. The final section of the paper is very confusing. The authors appear to suggest that the base model constrained to the measured formaldehyde overestimates the HO2/RO2 ratio by increasing the production of HO2 relative to RO2. However, unconstraining the model to the formaldehyde concentrations results in lower HO2/RO2 ratios that are in better agreement with the measured ratio, presumably because the model underestimates the measured formaldehyde. However, including monoterpene chemistry that have multiple RO2 isomerization steps increases the modeled RO2 concentration so that the modeled HO2/RO2 ratio is in better agreement with the measurements when HCHO is constrained. The authors suggest that additional measurements of OVOCs are necessary, but the connection between unmeasured OVOCs and the different model scenarios discussed in this section is not clear. This section of the manuscript needs considerable revision in order to clarify the points that the*

*authors are trying to make.*

**Reply:**

Thank you for your response. We acknowledge your point that the final section of the paper is very confusing. We wish to elaborate on a phenomenon in the manuscript, which is that formaldehyde has a high concentration distribution (average noon concentration of $21.81 \pm 4.57$ ppb), but OVOCs do not show the same intensity in contributing to ROx radicals as formaldehyde does. The current mechanism is having difficulty replicating the concentration of formaldehyde. Therefore, we have changed the title of the relevant section to "4.3 Missing OVOCs sources influence ozone production" and adjusted the content of that section. We have removed the impact of formaldehyde on the length of the reaction chain and its oxidizing effect, focusing more on the diagnostic of the $HO_2/RO_2$ ratio on ozone formation to improve the readability of the manuscript.

We also analyzed the impact of the missing OVOCs sources on $RO_2$ radicals and ozone production. When formaldehyde levels are unconstrained, the simulated $HO_2/RO_2$ ratios align with observations, suggesting that under the prevailing chemical mechanism, the photochemical efficiency of formaldehyde and other OVOCs is similar. Therefore, an empirical hypothesis is proposed to amplify the concentration of higher-order aldehydes by a factor of about 4, which is the proportion of formaldehyde concentration underestimated by the model. The qualitative assessment of the impact of missing aldehyde primary emissions on $RO_2$ radical concentrations was combined with the HAM mechanism across the entire photochemical spectrum (Fig.S12). Enhanced impact of aldehyde autoxidation in the presence of weak photochemical conditions could alter the simulated levels of OH and $HO_2$ radicals by approximately 13.9% and 18.1%, respectively. However, higher ALD concentrations will be achieved under intensive photochemical conditions, leading to the gradual dominance of the sink channels for OH + OVOCs, with the effect of autoxidation mechanisms gradually decreasing. $RO_2$ radical concentrations is notably more sensitive to the HAM mechanism, where incorporates additional OVOCs, can enhance the simulation of $RO_2$ radical concentrations by 20 - 40%.

We added the detailed description in Line 661-712.

[Figure]

**Fig. S12.** The relationship between the differences in the simulation of **(a)** OH, **(b)** $HO_2$, and **(c)** $RO_2$ radical concentrations by HAM mechanism and the base scenario across the entire photochemical spectrum. An empirical hypothesis is proposed to amplify the concentration of higher-order aldehydes by a factor of about 4, which is the proportion of formaldehyde concentration underestimated by the model. The boxplots represent the 10%, 25%, median, 75%, and 90% of the data, respectively.

**Revision:**

Line 660: 4.3 Missing OVOCs sources influence ozone production

Line 661-712: The consistency between model predictions and observed measurements for ozone production, akin to the concentration ratio of $HO_2/RO_2$, is depicted in Fig. 11(a)(b). In areas with low NO levels, the ratio of modeled to actual ozone production ranges from 0.5 to 2, with the exception of the ClearfLo and AIRPRO-summer datasets(Woodward-Massey et al., 2023; Whalley et al., 2021). Conversely, under high NO conditions (with NO concentrations between 3 and 6 ppbv), the ozone production rate (P(Ox)) derived from measured radical concentrations typically exceeds that of the base model's predictions by more than threefold. Laboratory experiments focusing on the oxidation of representative VOCs suggest that ozone production can be enhanced by approximately 25% for the anthropogenic VOCs under investigation(Färber et al., 2024). The MTS+X scenario

[revised manuscript text omitted]